# Dual Roles of Canagliflozin on Cholangiocarcinoma Cell Growth and Enhanced Growth Suppression in Combination with FK866

**DOI:** 10.3390/ijms26030978

**Published:** 2025-01-24

**Authors:** Daisuke Taguchi, Yohei Shirakami, Hiroyasu Sakai, Daisuke Minowa, Takao Miwa, Toshihide Maeda, Masaya Kubota, Kenji Imai, Takashi Ibuka, Masahito Shimizu

**Affiliations:** Department of Gastroenterology, Gifu University Graduate School of Medicine, Gifu 501-1194, Japan

**Keywords:** canagliflozin, cholangiocarcinoma, sirtuin 1, nicotinamide adenine dinucleotide+, nicotinamide phosphoribosyltransferase

## Abstract

Cholangiocarcinoma-associated mortality has been increasing over the past decade. The sodium-glucose cotransporter 2 inhibitor, canagliflozin, has demonstrated anti-tumor effects against several types of cancers; however, studies examining its potential impact on cholangiocarcinoma are lacking. This study investigated the anti-tumor effects of canagliflozin on cholangiocarcinoma and the effects of nicotinamide adenine dinucleotide (NAD)+ salvage pathway activation and sirtuin 1 on tumor growth. We evaluated cell proliferation and gene expression in several cholangiocarcinoma cell lines and analyzed the effects of canagliflozin on cell proliferation, apoptosis, and migration. Canagliflozin treatment decreased the viability of cholangiocarcinoma cells in a concentration-dependent manner but increased the viability at low concentrations in several cell lines. At high concentrations, canagliflozin arrested the cell cycle checkpoint in the G0/G1 phase. In contrast, at low concentrations, it increased the proportion of cells in the S phase. Canagliflozin also reduced the migratory ability of cholangiocarcinoma cells in a concentration-dependent manner. Canagliflozin treatment upregulated nicotinamide phosphoribosyltransferase (NAMPT), NAD+, and sirtuin 1 in cholangiocarcinoma and activated the NAD+ salvage pathway. The growth-inhibitory effect of canagliflozin was enhanced when combined with an NAMPT inhibitor. Canagliflozin inhibits cholangiocarcinoma cell growth and migration and its anti-tumor effect is enhanced when combined with an NAMPT inhibitor. However, further investigation is required because of its potential tumor growth-promoting effect through the activation of the NAD+ salvage pathway.

## 1. Introduction

Cholangiocarcinoma (CCA), the second most common primary liver cancer, is a highly heterogeneous malignancy that can occur in the intrahepatic, perihilar, or distal bile ducts and its mortality rates have increased over the past decade [1,2]. The use of chemotherapy for unresectable CCA has been gradual, with limited available regimens [3,4,5]. Recently, molecularly targeted drugs and immune checkpoint inhibitors have been introduced [6,7]. However, developing novel therapies for patients without targeted genomic alterations is imperative [8].

Sodium-glucose cotransporter 2 (SGLT2) inhibitors, including canagliflozin (CANA), have been developed for treating type 2 diabetes. They prevent the reabsorption of glucose, increasing its urinary excretion [9]. CANA has demonstrated favorable anti-tumor effects in vitro and growth suppression effects on xenografts, as well as improvements in some tumor-associated parameters in animal experiments for various types of cancers, including liver, pancreatic, breast, thyroid, stomach, and lung cancers [10,11,12,13,14,15,16,17,18]. The anti-tumor mechanism of CANA remains unclear, as multiple mechanisms other than SGLT2 inhibition have been speculated [13,17,18]. CANA reduces oxidative stress and improves energy metabolism by activating sirtuin 1 (SIRT1), a nicotinamide adenine dinucleotide (NAD)+-dependent class III histone deacetylase, in some non-tumor tissues [19,20,21]. The activation of SIRT1 has been associated with increased healthy life expectancy [22]. In contrast, SIRT1 activation promotes tumor growth and epithelial–mesenchymal transition (EMT) [23,24,25]. However, studies analyzing the efficacy of CANA in treating CCA or the effect of SIRT1 activation by CANA on cancer cells are lacking.

This present study reports the anti-cancer properties of CANA against human CCA cells. We demonstrate that CANA significantly inhibits cell growth of CCA with higher and lower expression of SGLT2 by the arresting cell cycle. This study also indicates that CANA inhibits cancer cell migration, which is the first reported mechanism. Notably, CANA exerts tumor-promoting effects through the NAD+ salvage pathway, which recycles nicotinamide into NAD+, and by activating SIRT1 at lower concentrations. Meanwhile, the anti-cancer effects of SGLT2 inhibitors have been the main focus of research. This finding indicates that CANA could have a negative effect on malignancy and may have dual roles on CCA cell growth. So far, no studies have evaluated the effects of CANA on SIRT1 and NAD+ salvage pathways in cancer cells. Here, we report for the first time that CANA treatment activates the NAD+ pathway and upregulates SIRT1 expression in CCA cells.

## 2. Results

### 2.1. Gene Expression of SGLT1 and SGLT2 in CCA Cells

The expression of genes encoding SGLT1 and SGLT2 in each CCA cell line was evaluated and compared with that in previously reported hepatocellular carcinoma (HCC) cell lines [10]. According to the report, the HCC cell line HLE was used as a control for the low expression of *SLC5A1* and *SLC5A2*, and Huh7 was used as a control for the high expression of *SLC5A1* and as a reference for *SLC5A2* expression in this study. The gene expression of *SLC5A1* was significantly higher in HuCCT1 cells than in HLE cells and significantly lower in Huh28 cells than in Huh7 cells (Figure 1A). The gene expression of *SLC5A2* was significantly lower in TFK-1 cells than in Huh7 cells and significantly higher in Huh28 cells than in HLE cells (Figure 1B).

A comparison of CCA cells and the surrounding normal tissues using the CCA dataset from the Cancer Genome Atlas (TCGA) showed that *SLC5A1*- and *SLC5A2*-encoding genes were highly expressed in CCA tissues (Figure 1C).

### 2.2. CANA Affects the Growth and Survival of CCA Cell Lines

Cell viability assessment using a 3-(4,5-dimethylthiazol-2-yl)-5-(3-carboxymethoxyphenyl)-2-(4-sulfophenyl)-2H-tetrazolium (MTS) assay showed that CANA induced a concentration-dependent decrease in the survival of HuCCT1, Huh28, and TFK-1 with median inhibition concentration (IC_50_) and 95% confidence interval of 52.9 µM (50.6–55.2), 42.6 µM (40.2–45.1), and 46.1 µM (44.4–47.7), respectively. HuCCT1 showed a substantial increase in survival at 30 µM and TFK-1 showed improved survival at lower doses (Figure 1D). In addition, we referred to the previous study to determine the concentrations of CANA for treating CCA cells [10]. We treated CCA cells with similar concentrations of CANA, performed a cell proliferation assay, and determined the IC_50_ values for each CCA cell. Notably, we discovered that a relatively low dose (30 μM) of CANA treatment promoted cell proliferation in HuCCT1. Therefore, the following concentrations of CANA were used for HuCCT1 in subsequent experiments: 30 µM, 50 µM, and 80 µM. At 30 μM, CANA promoted the cell proliferation of HuCCT1 as described above, 50 µM was the IC_50_ of HuCCT1 cells, and 80 µM was the maximum concentration that could be adjusted, considering the effect of the solvent dimethyl sulfoxide.

### 2.3. CANA Arrests the Cell Cycle of CCA Cells

We evaluated the gene expression of cell cycle-related genes to determine the effect of each CANA concentration on the cell cycle of HuCCT1 cells. We discovered a concentration-dependent increase in *CDKN1A*, the P21 coding gene, and a concomitant decrease in *CDK1* and *CDK2*, which encode cyclin-dependent kinases 1 and 2, respectively (Figure 2A). *CCNB1*, a cyclin B1 coding gene, also decreased in a concentration-dependent manner, whereas *CCNE1*, a cyclin E1 coding gene, was not significantly affected, and *CCND1*, a cyclin D1, showed a concentration-dependent increase (Figure 2B). The gene expression of *MKI67*, which encodes the marker of proliferation, Ki-67, which reflects the cell cycle outside the G0 phase, decreased significantly in a concentration-dependent manner (Figure 2C).

The cell cycle rate of the HuCCT1 cells was assessed using a cell cycle assay (Figure 3A). Treatment with 80 μM CANA significantly decreased the S and G2/M phases and increased the G0/G1 phase, whereas treatment with 30 μM CANA significantly increased the S phase (Figure 3B).

### 2.4. Effect of CANA on Apoptosis Is Limited in CCA Cells

Treatment with CANA did not affect the expression of *BAX*, which encodes the apoptotic regulator B-cell lymphoma (Bcl)-associated protein (BAX). However, it significantly increased the expression of *BCL2* and *BCL2L1*, which encode the apoptotic regulators Bcl-2 and Bcl-2-like protein 1, respectively, in a concentration-dependent manner (Figure 4A). HuCCT1 was treated with 50 µM of CANA for 0–24 h to access actual apoptosis (Figure 4B). Compared with the controls, the percentage of cells in the Q2 region, which defines apoptosis and cell death, was unchanged after 24 h and the percentage of cells in the Q4 region, which defines early apoptosis, showed a significant increase after 24 h. However, the increase was negligible (2%) (Figure 4C). The percentage of cells in the Q4 region did not change significantly when treated with 80 µM CANA for 24 h (Figure 4D).

### 2.5. CANA Suppresses EMT in CCA

The expression of EMT-related genes in HuCCT1 cells was evaluated after treatment with various concentrations of CANA. *CDH1* and *TJP1*, which encode the epithelial marker E-cadherin and tight junction protein ZO-1, respectively, were significantly upregulated at high concentrations of CANA (Figure 5A). *CDH2*, which encodes the mesenchymal marker N-cadherin, showed a concentration-dependent decrease in expression. In contrast, other mesenchymal marker genes showed increased expression at high concentrations (Figure 5B).

A wound-healing assay was performed to assess the migration ability, which demonstrated a CANA concentration-dependent delay in wound healing in HuCCT1 cells (Figure 5C). After treatment with CANA for 20 h, wound healing was significantly delayed at concentrations of 50 µM and 80 µM, with a wound area of 33.0 ± 2.9% and 72.5 ± 4.5%, respectively. In addition, complete wound closure was observed after 40 h of treatment with CANA at concentrations of 0 µM, 30 µM, and 50 µM, whereas CANA 80 µM left 33.2 ± 8.3% of the wounds open, indicating a significant delay in wound healing (Figure 5D).

### 2.6. Effects of CANA on Gene Expression of Its Target Proteins

CANA inhibited SGLT1 and SGLT2, but in a concentration-dependent increase in the expression of *SLC5A1* and *SLC5A2* (Figure 6A). Histone deacetylase 6 (HDAC6), a member of the sirtuin family, is a direct target of CANA [18]. CANA significantly reduced *HDAC6* expression in a concentration-dependent manner (Figure 6B).

### 2.7. CANA Acts on the NAD+ Salvage Pathway and Promotes Cell Growth

We evaluated the effects of CANA on nicotinamide phosphoribosyltransferase (NAMPT), the rate-limiting enzyme in the NAD+ salvage pathway, and SIRT1, an NAD+-dependent deacetylase. CANA increased the expression of NAMPT and SIRT1 in a concentration-dependent manner (Figure 6C). The NAD+/NADH assay was performed to measure intracellular NAD+ levels and showed a concentration-dependent increase in NAD+ levels, although not significant, indicating the activation of the NAD+ salvage pathway (Figure 6D). To evaluate how the activation of the NAD+ salvage pathway affects tumor growth, we evaluated the cell viability of the NAMPT inhibitor FK866 in combination with CANA to suppress the NAMPT-NAD+-SIRT1 pathway. Compared with CANA alone, the combination of CANA and FK866 significantly decreased cell viability, suggesting that activation of the NAD+ salvage pathway promotes tumor growth (Figure 6E).

## 3. Discussion

A novel finding of this study is that CANA might have dual roles on CCA cell growth. Our findings demonstrate that CANA inhibits cell proliferation and EMT in CCA cell lines; however, the activation of SIRT1 by CANA through the NAD+ salvage pathway can promote the proliferation of CCA cells. These findings highlight the enhanced anti-tumor effect of CANA when combined with NAMPT inhibitors, which suppress the NAD+ salvage pathway.

High concentrations of CANA significantly reduced CCA cell viability. SGLT2 inhibition by CANA has anti-tumor effects outside of CCA [10,12,14,15,16,17]. However, given that CANA exerts anti-tumor effects not mediated by SGLT2 [13,18], the mechanism of action of CANA remains unclear. The gene expression of *SLC5A1* and *SLC5A2* and the protein expression of SGLT1 and SGLT2 have increased in several cancer cell lines and human cancer tissues [10,12,15]. In our analysis of the TCGA database, we found that *SLC5A1* and *SLC5A2* gene expression increased in CCA, consistent with the findings observed in other types of cancers. However, each cell line had similar IC_50_ values when treated with CANA despite considerable differences in the gene expression of *SLC5A2* in CCA cell lines. This finding also suggests that the anti-cancer effect of CANA on CCA cells is only partially dependent on SGLT2. Dapagliflozin, an SGLT2 inhibitor, has anti-tumor effects [12,13,15], whereas tofogliflozin, a more selective SGLT2 inhibitor, has no direct effect on tumor growth inhibition [26]. In addition, additional anti-tumor effects of CANA have been reported in SGLT2 knockdown cells [13]. CANA binds specifically to HDAC6 and directly inhibits HDAC6 [18]. Furthermore, HDAC6 inhibition reduces CCA cell growth by restoring primary cilia [27]. These reports support our finding that CANA reduces *HDAC6* gene expression in CCA in a concentration-dependent manner and inhibits tumor growth. Our findings demonstrate that, in addition to inhibiting SGLTs, multiple mechanisms are involved in the anti-tumor effect of CANA.

To date, most papers examining the anti-cancer effects of SGLT2 inhibitors have focused on SGLT2 expression in cancer cells. That is, the inhibitors are presumably effective only on SGLT2-expressing cancer cells [10,28]. Considered to be of clinical relevance, CANA may be used in the future for selected patients with tumors expressing SGLT2 as a biomarker based on the results of a companion diagnosis. A previous study by Scafoglio et al. [28] indicated that SGLT2 is functionally expressed in patients with pancreatic cancer and SLGT2 inhibitors might be useful for cancer therapy. However, our present study, as well as previous reports, indicated that CANA has anti-tumor effects regardless of SGLT2 expression in the cells [13,27]. Since the anti-cancer effects of SGLT2 inhibitors may depend on the type of cancer and/or the type of SGLT2 inhibitor, further research focusing on the dependence and difference described above may be able to clarify detailed mechanisms. The effect of glucagon-like peptide (GLP)-1 agonist, another anti-diabetic agent, on malignancy has also been reported [29]. Similar to CANA in this study, GLP-1 agonists exhibit anti-cancer effects as well as tumor-promoting effects which appear to depend on the cancer type [29,30]. No report has compared the anti-cancer effects of SGLT2 inhibitors and GLP-1 agonists. However, a retrospective cohort study indicates that the group treated with SGLT2 inhibitors had a lower incidence of cancers than the group treated with dipeptidyl peptidase-4 inhibitors which are known to increase blood GLP-1 levels [31].

Our findings demonstrated that treatment with high concentrations of CANA reduced the percentage of cells in the S and G2/M phases and increased the percentage of cells in the G0/G1 phase of the cell cycle in a CCA cell line. This is consistent with those of previous studies, which reported that CANA causes G1/S arrest [15]. The significant decrease in Ki-67 suggests that the anti-tumor effect of CCA is mainly due to cell cycle arrest. However, its effects on apoptosis are limited. CCA cell lines treated with CANA showed an approximately 2% increase in early apoptosis but no effect on late apoptosis or cell death. However, conflicting results indicating that CANA induces apoptosis in several types of cancer have been reported. Apoptosis was induced in liver and thyroid cancer cell lines at concentrations as low as 10 µM of CANA [10,15]; however, higher concentrations > 60–200 µM are required to induce apoptosis in pancreatic cancer cell lines [11,32]. The effect of CANA on apoptosis varies depending on the cancer type. Apoptosis may be induced in CCA at higher concentrations of CANA than those used in our study; however, the effect of CANA concentration at IC_50_ on apoptosis was limited in CCA.

NAD+ is a key coenzyme involved in essential physiological functions, including energy metabolism, DNA repair, and cell growth. The NAD+ salvage pathway is known as the two-step process for recycling nicotinamide into NAD+, where NAMPT converts nicotinamide into nicotinamide mononucleotide, which is then converted into NAD+ by nicotinamide/nicotinic acid mononucleotide adenylyltransferase. These processes are often dysregulated in cancer cells. Hence, the NAD+ salvage pathway is considered a promising target for cancer treatment strategies [33]. Pharmacological inhibition of this pathway induces cancer cell cytotoxicity by depleting energy levels, increasing sensitivity to oxidative damage, and disrupting cell signaling pathways such as SIRT1 and p53. A previous paper reported that inhibiting SIRT1, an NAD+-dependent enzyme, induces cyclin D1 downregulation and suppresses tumor growth in HCC cells [34]. In contrast, SIRT1 activation inhibits cyclin D1 transcription and cell growth in gastric cancer [35], suggesting two aspects of SIRT1 activation in promoting or suppressing tumor growth. In cells other than cancer cells, SIRT1 activation has beneficial effects on cardiovascular and hepatic cells, thereby extending life expectancy [22,36]. Regarding the relationship between CANA and SIRT1, it has been reported that SIRT1 is upregulated by CANA, resulting in decreased oxidative stress and improved energy metabolism, thereby leading to cardiovascular protection and the amelioration of ulcerative colitis [19,20,21]. However, no studies have evaluated the effects of CANA on SIRT1 and NAD+ pathways in cancer cells. Here, we report for the first time that CANA treatment activates the NAD+ salvage pathway and upregulates SIRT1 expression in CCA cells. An NAMPT inhibitor, which suppresses the NAD+ salvage pathway, was used in combination with CANA to determine the effect of CANA-induced activation of the NAD+ salvage pathway and SIRT1 in CCA [33]. Surprisingly, the inhibitor enhanced the anti-tumor effect of CANA by suppressing the NAD+ salvage pathway. The results suggest that the activation of the NAD+ salvage pathway and SIRT1 by CANA might promote tumor growth, which is consistent with those of previous studies reporting that SIRT1 suppression inhibits CCA growth [27].

To date, the potential effects of CANA in promoting cancer cell growth have not been fully investigated. However, a previous study showed that CANA increased polyps in a mouse colon polyposis model [37], suggesting that CANA may have potential tumor-promoting effects. This present study also demonstrated that CANA increased the survival of CCA cells, especially at low concentrations. In our study, CANA treatment increased SGLT gene expression. In contrast, previous studies have demonstrated a decrease in glucose uptake due to the natural action of CANA [10,15]. Therefore, we speculate that the elevated SGLT levels may be secondary to upward regulatory feedback in SGLTs’ gene expression to counteract the decreased glucose uptake. Moreover, activation of the NAD+ salvage pathway promotes tumor growth and leads to increased survival.

In this study, CANA inhibited EMT in CCA cells. The inhibitory effect of CANA on the migratory ability of cancer cells has also been reported previously [17,18]. Our findings demonstrate that the expression of some mesenchymal markers was elevated in CCA cells, which is consistent with the findings of previous studies elucidating that SIRT1 activation elevates the expression of mesenchymal markers [24,25]. In contrast, CANA treatment increased the expression of epithelial markers and decreased the expression of N-cadherin. Furthermore, CANA delayed wound healing in CCA cells, demonstrating that CANA inhibited EMT in CCA in a concentration-dependent manner.

Our study had several limitations. First, CANA could only be studied in vitro and the IC_50_ concentrations of CANA obtained from the in vitro studies were 5–15-fold higher than the actual blood concentrations after regular doses of CANA intake in humans. However, CANA has been extensively investigated in other cancer cells in xenograft and in vivo carcinogenesis models, where the CANA concentrations used were 50–100 times higher than those used in clinical settings. The dose of CANA used in these studies consistently showed good anti-tumor effects with fewer adverse events such as hypoglycemia and weight loss [10,11,15]. In addition, there is a report of anti-tumor effects even at lower concentrations of CANA than the IC_50_ calculated in vitro for liver cancer if administered for a prolonged period [10], suggesting that the serum concentration may not be needed to reach the level required in vitro because the drug is administered for a longer period. Moreover, there are several epidemiological studies investigating the incidence of malignancies in patients treated with SGLT2 inhibitors, showing that the incidence was lower in the treated group [31,38,39]. From the results of these studies, we speculate there is a possibility that SGLT2 inhibitors can show anti-tumor effects for CCA in animal and human studies. The tumor-promoting effect of CANA at low concentrations observed in our study may hinder in vivo animal examinations as a next step for its application in cancer chemotherapy; however, this may be overcome by agents that inhibit the NAD+ salvage pathway, such as NAMPT inhibitors. In a xenograft model of pancreatic cancer, it was reported that SGLT2 inhibitors blocked glucose uptake and reduced tumor growth and survival [28]. Based on the results of the present study, animal CCA models or patient-derived xenograft studies would provide further validation of the dual effects of CANA. Second, we did not investigate protein levels but mainly gene expressions of major molecules in this study. In addition, a multifaceted evaluation was not conducted for apoptosis and migration/invasion. Therefore, a more detailed analysis of protein expression, apoptosis, and migration/invasion would provide a more comprehensive understanding of the effects of CANA against malignancy. Third, CANA treatment-induced metabolic changes and compensatory alterations of other pathways, such as glucose transporter type 1 upregulation or glycolysis, were not investigated in this study. A previous study demonstrated that CANA decreased glucose uptake in cancer cells in a dose-dependent manner [10], suggesting that the cancer cells, at least in part, depend on SGLT2 for glucose uptake and compensatory mechanisms for glucose uptake may not work after CANA treatment. In the present study, SGLT2 gene expression was upregulated by the treatment with CANA. This upregulation might be a compensatory mechanism of CCA cells. That is. when CCA cells are exposed to SGLT2 inhibitors, the cells might try to respond by further overexpressing the transporter for glucose uptake. However, the SGLT2 activity and glucose uptake have not been examined. Further research is required to reveal how cell viability is increased and the S phase is progressed by low-dose treatment of CANA.

In conclusion, CANA inhibited tumor growth and EMT by inducing cell cycle arrest in CCA cells and its anti-tumor effect was enhanced when combined with NAMPT inhibitors. In addition, it may also exert tumor-promoting effects through the NAD+ salvage pathway and SIRT1 activation. Much attention has been focused solely on the anti-tumor effects of SGLT2 inhibitors, including CANA; however, we suggest that the opposite effect should also be noted. Further studies, including animal CCA models or CCA xenograft examinations for the next step, are warranted to investigate the dual effects of CANA in CCA.

## 4. Materials and Methods

### 4.1. Cell Culture and Reagents

The CCA cell lines HuCCT1 (RRID: CVCL_0324) and Huh28 (RRID: CVCL_2955) and the HCC cell lines HLE (RRID: CVCL_1281) and Huh7 (RRID: CVCL_0336) were purchased from Japanese Collection of Research Bioresources Cell Bank (Osaka, Japan). The CCA cell line TFK-1 (RRID: CVCL_2214) was purchased from RIKEN BioResource Research Center (Ibaraki, Japan). The cell lines were cultured in Roswell Park Memorial Institute-1640 medium (FUJIFILM Wako Pure Chemical Corporation, Osaka, Japan) supplemented with 10% fetal bovine serum (Sigma-Aldrich, St. Louis, MO, USA) and 1% penicillin–streptomycin (Thermo Fisher Scientific Inc., Waltham, MA, USA). Cell cultures were incubated at 37 °C in a humidified atmosphere containing 5% CO_2_. All experiments were performed with mycoplasma-free cells. After all experiments were completed, all cell lines were authenticated using STR profiling (Japanese Collection of Research Bioresources Cell Bank, Osaka, Japan).

CANA (MedChemExpress, Monmouth Junction, NJ, USA) and the NAMPT inhibitor, FK866 (Selleck Chemicals, Houston, TX, USA), were adjusted throughout the experiment to ensure that the concentration of dimethyl sulfoxide in the experimental solutions did not exceed ≤0.1% at each concentration.

### 4.2. Cell Viability

MTS assays were performed using the CellTiter 96 AQueous One Solution Cell Proliferation Assay (Promega Corporation, Madison, WI, USA) to evaluate the effect of each drug on the viability of CCA cell lines. In the present study, we referred to a previous study to determine the concentrations of CANA for treating CCA cells [10]. CCA cell lines of 2.5 × 10^3^ per well were seeded into 96-well plates with 8 wells in each group and treated with CANA at concentrations of 0–80 μM for 48 h. The same treatment was performed using the CANA combined with FK866. Subsequently, 20 μL of MTS solution was added to each well and incubated for 2 h at 37 °C in a humidified atmosphere with 5% CO_2_. The absorbance was measured using a Multiskan FC (Thermo Fisher Scientific Inc., Waltham, MA, USA). The absorbance of each sample minus the absorbance of the blank was calculated and the percentage of each sample relative to the average of the controls was defined as cell viability.

### 4.3. Gene Expression

Total RNA extraction, complementary DNA synthesis in CCA and HCC cell lines, and real-time quantitative reverse transcription polymerase chain reaction analysis were performed using previously described methods [40]. The glyceraldehyde-3-phosphate dehydrogenase coding GAPDH, amplified in parallel, was used as the housekeeping gene. The primer sequences used to amplify specific genes are summarized in supporting information (Table 1).

### 4.4. Database Analysis

The TCGA CCA dataset was analyzed using the UALCAN portal [41,42], a web source available for analyzing gene expression data. The expression of *SLC5A1*, the SGLT1 coding gene, and *SLC5A2*, the SGLT2 coding gene, in CCA was examined in contrast to the surrounding normal tissue.

### 4.5. Wound-Healing Assay

A wound-healing assay was performed to evaluate the effect of CANA on the migration potential of CCA cells. HuCCT1 cells were seeded in 6-well plates and cultured in a medium until the monolayer was 90–95% confluent. The cells were incubated in the medium containing mitomycin C (FUJIFILM Wako Pure Chemical Corporation, Osaka, Japan) (10 µg/mL) at 37 °C and 5% CO_2_ for 1 h. After incubation, the medium containing mitomycin C was aspirated. The wells were washed once with phosphate-buffered saline (PBS) and aspirated. The monolayer was scratched with a 1000 µL tip, fresh PBS (3 mL/well) was added to the wells, and the cells were washed and aspirated three times with PBS. The wound was photographed at 40x in the absence of medium and PBS and the cells were then treated with various concentrations of CANA for 40 h. Photographs were captured after 20 and 40 h without medium or PBS. The wounded areas were measured using ImageJ version 1.54g software (National Institutes of Health, Bethesda, MD, USA). The images were displayed in 8-bit grayscale.

### 4.6. Cell Cycle Assay

Cell cycle assays were performed using the Cell Cycle Assay Solution Deep Red (DOJINDO LABORATORIES, Kumamoto, Japan) with some modifications to the manufacturer’s protocol. Specifically, pellets of 5 × 10^5^ HuCCT1 cells treated with each concentration of CANA or solvent were prepared, 1 mL of cold 70% ethanol was added, and cells were incubated at 4 °C for 2 h for fixation. After centrifugation at 3000× *g* (modifications) for 5 min to remove alcohol, the cells were washed once with 1000 µL of PBS and suspended in 500 µL of PBS. Subsequently, 5 μL of Cell Cycle Assay Solution was added to the cell suspension, mixed using a vortex mixer, and incubated at 37 °C for 15 min under light-shielded conditions. Then, the sample was passed through a cell strainer for flow cytometry measurement. The BD FACSCanto II flow cytometer (BD Biosciences, San Jose, CA, USA) (excitation = 633 nm; emission = 780/60 nm BP) was used and analysis was performed using BD FACSDiva version 6.1.3 software (BD Biosciences, San Jose, CA, USA).

### 4.7. Apoptosis Assay

An apoptosis assay was performed using the BD Pharmingen FITC Annexin V Apoptosis Detection Kit I (BD Biosciences, San Jose, CA, USA) to evaluate whether CANA induces apoptosis. HuCCT1 cells were treated with 6 µM of camptothecin (FUJIFILM Wako Pure Chemical Corporation, Osaka, Japan) for 4 h to induce apoptosis and used as controls. Apoptosis-induced HuCCT1 cells were divided into untreated, FITC-labeled annexin V-treated, propidium iodide (PI)-treated, and Annexin V and PI co-treated groups as measured by flow cytometry. Excitation = 488 nm and emission = 530/30 nm BP for FITC-labeled annexin V-positive cells. Excitation, 488 nm and emission, 670 nm for PI-positive cells. The compensation was adjusted such that each condition was distributed at defined locations (PI-%FITC: 31.8; FITC-%PI: 0.0). HuCCT1 cells treated with each concentration of CANA for 0–24 h were co-stained with annexin V and PI and measured using BD FACSCanto II under the conditions described above. The results were analyzed using BD FACSDiva software.

### 4.8. NAD+/NADH Assay

Quantification of intracellular NAD+/NADH was performed using the NAD/NADH Assay Kit-WST (DOJINDO LABORATORIES, Kumamoto, Japan) according to the manufacturer’s protocol. Pellets of 10 × 10^5^ HuCCT1 cell lines treated with each CANA concentration were prepared in microtubes, washed with PBS, lysed with reagent, and filtered by centrifugation. The filtrate was divided into two groups and stored in an ice bath to determine the total NAD+/NADH levels or heat-treated at 60 °C for 60 min to determine the NADH level, and then reacted with the working solution in 96-well culture plates at 37 °C for 60 min. The absorbance at 450 nm was measured using a plate reader and the average was calculated. The total NAD+/NADH and NADH levels were calculated from the dosimeters using standard solutions measured simultaneously and standardized by cell counting. NAD+ levels were calculated by subtracting NADH levels from the total NAD+/NADH. All data were obtained in triplicate from three independent experiments.

### 4.9. Statistical Analysis

All data are presented as mean ± standard error of the mean or mean ± confidence intervals. Unless otherwise stated, the experiments were conducted in parallel (*n* = 6 in each group). Nonparametric statistical analyses between three or more groups were performed using the Kruskal–Wallis test, followed by the Steel–Dwass test. A two-sided *p*-value < 0.05 indicated statistical significance. All statistical analyses were performed using the R version 4.3.1 software (The R Foundation for Statistical Computing, Vienna, Austria). The IC_50_ and two-sided 95% confidence interval were calculated using the drc package in the R software.

## Figures and Tables

**Figure 1 ijms-26-00978-f001:**
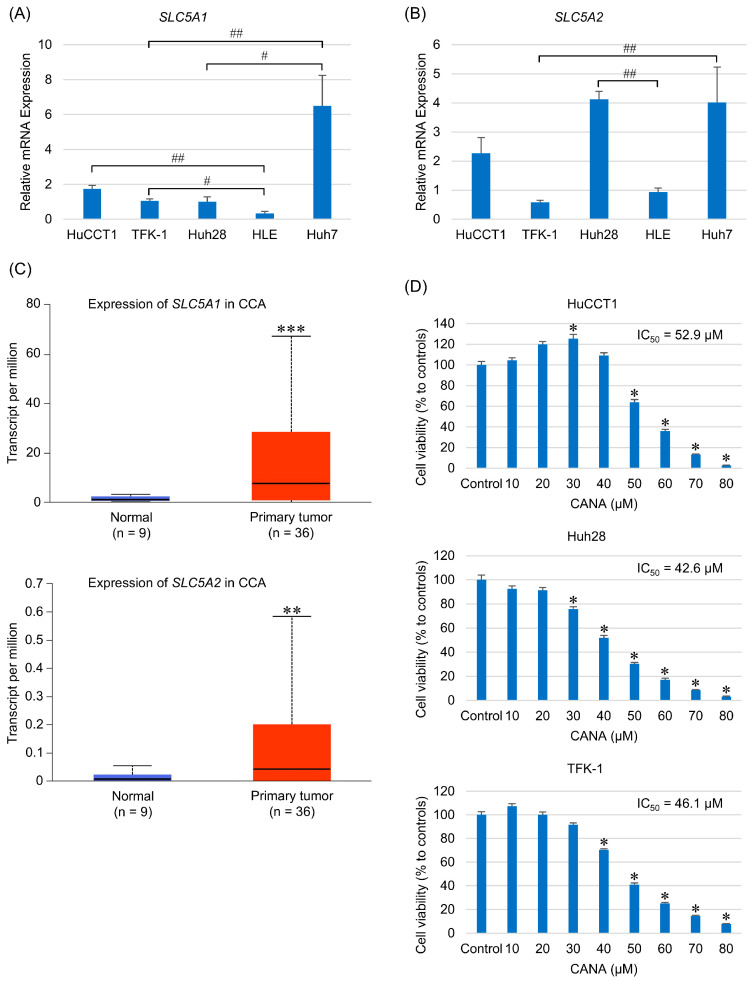
Gene expression of SGLTs in CCA cells and CANA-induced changes in CCA viability. (**A**) *SLC5A1* and (**B**) *SLC5A2* levels in CCA cell lines were validated by qRT-PCR and compared with those in HCC cell lines (*n* = 8). (**C**) *SLC5A1* and *SLC5A2* expression in CCA tissues from the TCGA database was analyzed in comparison with adjacent normal tissues. (**D**) HuCCT1, Huh28, and TFK-1 cells were examined for viability at various concentrations of CANA using a cell proliferation assay (*n* = 8). * *p* < 0.05, ** *p* < 0.01, and *** *p* < 0.005 compared with the control group; ^#^
*p* < 0.05 and ^##^
*p* < 0.01 compared between each group. Abbreviations: CANA, canagliflozin; CCA, cholangiocarcinoma; HCC, hepatocellular carcinoma; IC_50_, median inhibition concentration; qRT-PCR, quantitative reverse transcription polymerase chain reaction; SGLTs, sodium-glucose cotransporters; TCGA, the Cancer Genome Atlas.

**Figure 2 ijms-26-00978-f002:**
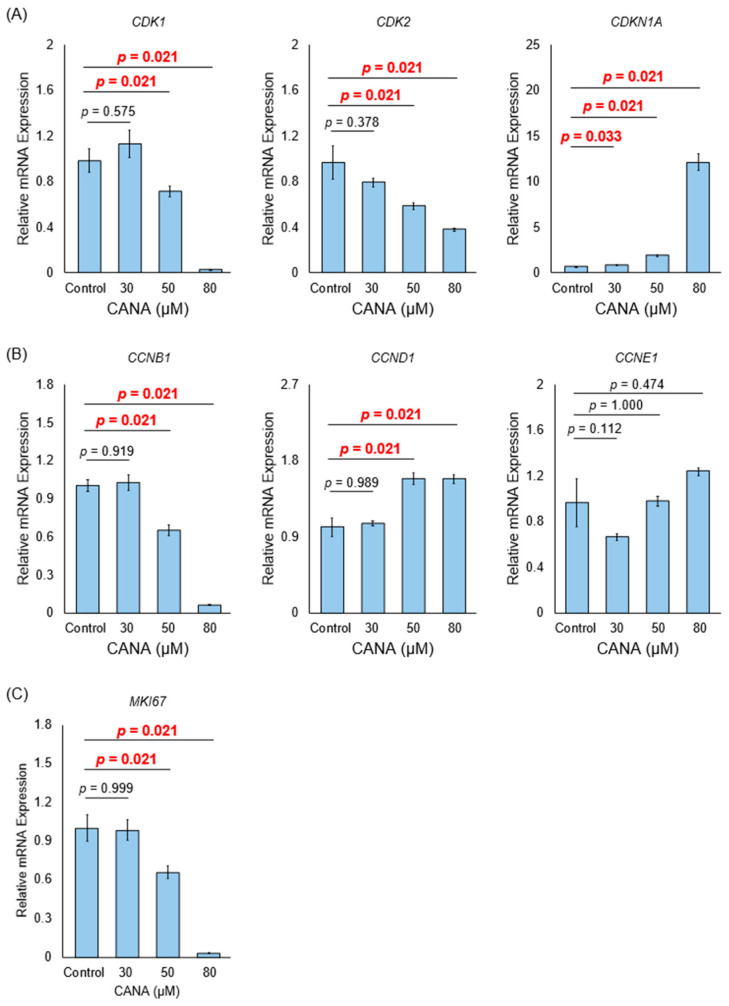
Effects of CANA on cell cycle checkpoints and cell proliferation markers in CCA cells. The mRNA expressions of (**A**) *CDK1*, *CDK2*, and *CDKN1A*, (**B**) *CCNB1*, *CCND1*, and *CCNE1*, and (**C**) *MKI67* in HuCCT1 cells after treatment with increasing concentrations of CANA were verified using qRT-PCR (*n* = 6). Data are presented as mean ± confidence intervals. Abbreviations: CANA, canagliflozin; CCA, cholangiocarcinoma; qRT-PCR, quantitative reverse transcription polymerase chain reaction.

**Figure 3 ijms-26-00978-f003:**
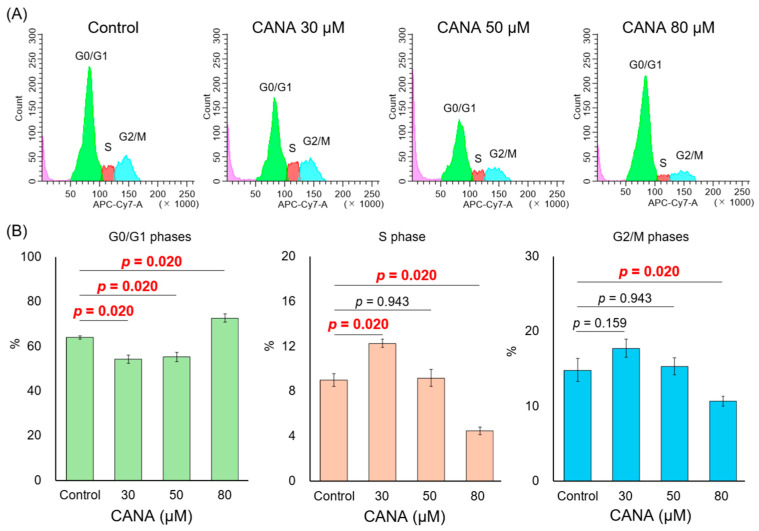
Effect of CANA on the cell cycle in CCA cells. (**A**) The cell cycle of HuCCT1 cells after treatment with several graded concentrations of CANA was verified by a cell cycle assay. The histograms colored green, red, and blue represent the G0/G1, S, and G2/M phases, respectively. (**B**) The percentages of cells in the G0/G1, S, and G2/M phases were quantified. (*n* = 6). Data are presented as mean ± confidence intervals. Abbreviations: CANA, canagliflozin; CCA, cholangiocarcinoma.

**Figure 4 ijms-26-00978-f004:**
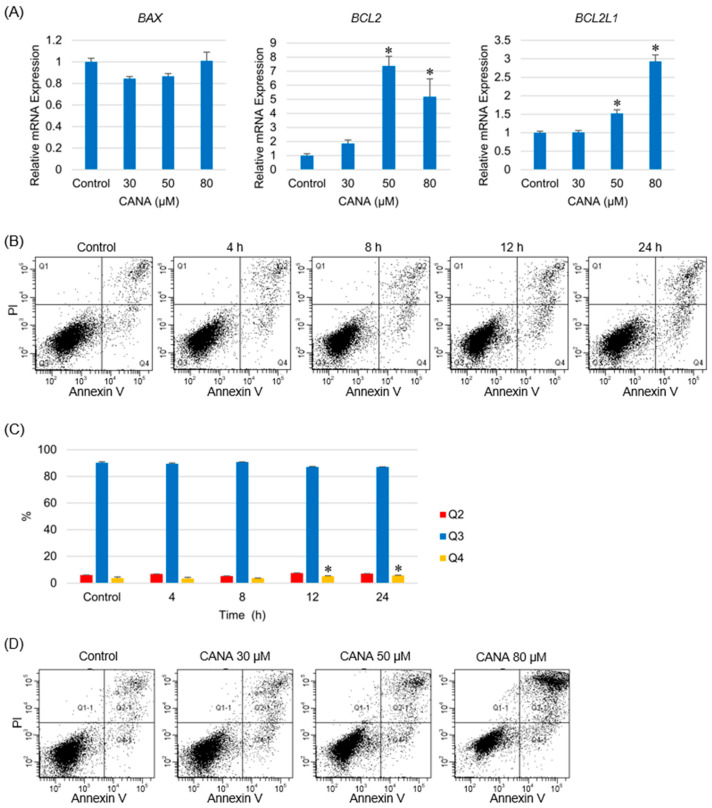
Effect of CANA on apoptosis in CCA cells. (**A**) The mRNA expressions of *BAX*, *BCL2*, and *BCL2L1* in HuCCT1 cells after treatment with several graded concentrations of CANA were verified by qRT-PCR (*n* = 6). (**B**) Apoptosis of HuCCT1 cells at several time points after treatment with 50 µM CANA was verified using the Annexin V apoptosis assay. (**C**) The percentage of cells in the Q2 (late apoptotic cells and dead cells), Q3 (non-apoptotic cells), and Q4 (early apoptotic cells) regions was quantified. (*n* = 6). (**D**) Apoptosis of HuCCT1 cells treated with several concentrations of CANA for 24 h was verified using the Annexin V apoptosis assay. * *p* < 0.05 compared with the control group. Abbreviations: CANA, canagliflozin; CCA, cholangiocarcinoma; PI, propidium iodide; qRT-PCR, quantitative reverse transcription polymerase chain reaction.

**Figure 5 ijms-26-00978-f005:**
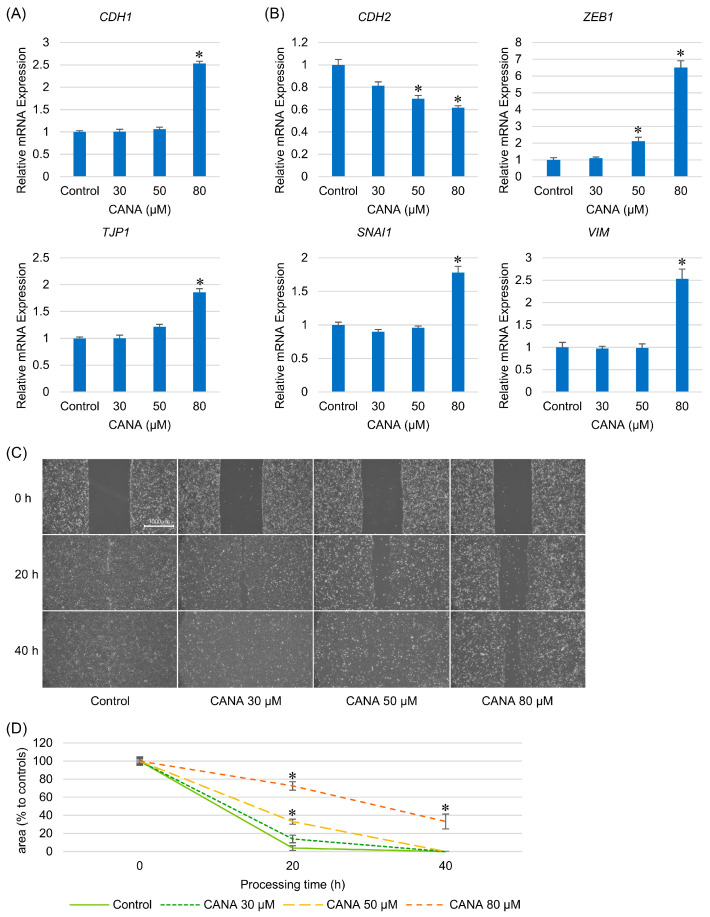
Effects of CANA on epithelial–mesenchymal transition markers and cellular migration in CCA cells. (**A**) The mRNA expressions of *CDH1* and *TJP1* as epithelial markers in HuCCT1 cells after treatment with several graded concentrations of CANA were verified by qRT-PCR (*n* = 6). (**B**) The mRNA expressions of *CDH2*, *ZEB1*, *SNAI1*, and *VIM* as mesenchymal markers in HuCCT1 cells after treatment with several graded concentrations of CANA were verified by qRT-PCR (*n* = 6). (**C**) Proliferation-suppressed monolayer fully confluent HuCCT1 cells were wounded by scratching and the wounds were photographed at several time points after treatment with several graded concentrations of CANA. (**D**) The wounded areas were quantified using ImageJ software (*n* = 6). * *p* < 0.05 compared with the control group. Abbreviations: CANA, canagliflozin; CCA, cholangiocarcinoma; qRT-PCR, quantitative reverse transcription polymerase chain reaction.

**Figure 6 ijms-26-00978-f006:**
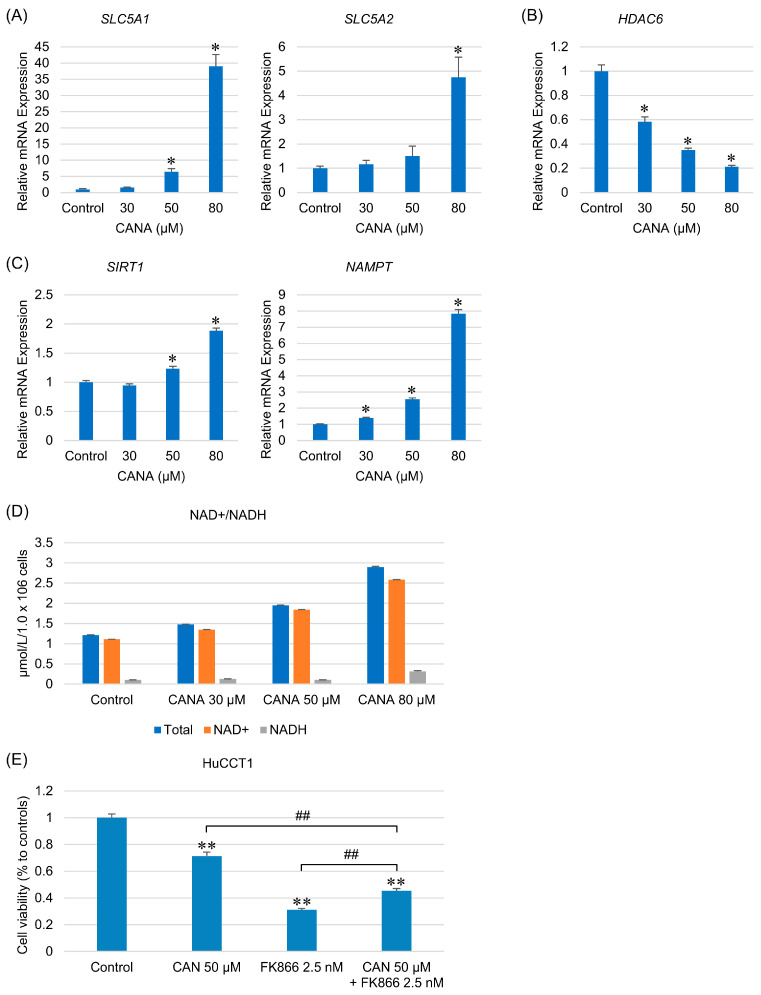
Effects of CANA on target proteins and NAD+ salvage pathways in CCA cells. The mRNA expressions of (**A**) *SLC5A1*, *SLC5A2*, (**B**) *HDAC6*, (**C**) *SIRT1*, and *NAMPT* in HuCCT1 cells after treatment with several graded concentrations of CANA for 48 h were verified by qRT-PCR (*n* = 6). (**D**) The NAD+/NADH levels in HuCCT1 cells after treatment with several graded concentrations of CANA were measured by the NAD+/NADH assay (*n* = 3). (**E**) HuCCT1 cells were examined for viability at 50 µM CANA alone, 2.5 nM FK866 alone, and combination treatment with CANA and FK866 in a cell proliferation assay (*n* = 8). * *p* < 0.05 and ** *p* < 0.01 compared with the control group; ^##^
*p* < 0.01 compared between each group. Abbreviations: CANA, canagliflozin; CCA, cholangiocarcinoma; NAMPT, nicotinamide phosphoribosyltransferase; qRT-PCR, quantitative reverse transcription polymerase chain reaction.

**Table 1 ijms-26-00978-t001:** Primer sequences.

Gene	Forward	Reverse
* BAX *	CATCATGGGCTGGACATTG	GGGACATCAGTCGCTTCAGT
* BCL2 *	AGTACCTGAACCGGCACCT	GCCGTACAGTTCCACAAAGG
* BCL2L1 *	GCCACTTACCTGAATGACCAC	TGCTGCATTGTTCCCATAGA
* CCNB1 *	CATGGTGCACTTTCCTCCTT	AGGTAATGTTGTAGAGTTGGTGTCC
* CCND1 *	CCATCCAGTGGAGGTTTGTC	GTGGGACAGGTGGCCTTT
* CCNE1 *	GGGACACCATGAAGGAGGA	TCTTCATCTGGATCCTGCAA
* CDH1 *	TGGAGGAATTCTTGCTTTGC	CGCTCTCCTCCGAAGAAAC
* CDH2 *	AGGCTTCTGGTGAAATCGCA	AGAGGCTGTCCTTCATGCAC
* CDK1 *	TGGATCTGAAGAAATACTTGGATTCTA	CAATCCCCTGTAGGATTTGG
* CDK2 *	AAAGCCAGAAACAAGTTGACG	GTACTGGGCACACCCTCAGT
* CDKN1A *	TGGGTGGTACCCTCTGGA	TGAATTTCATAACCGCCTGTG
* GAPDH *	AGCCACATCGCTCAGACAC	GCCCAATACGACCAAATCC
* HDAC6 *	GTCGCGGGGAAAAGGTCG	CTCCACTCATTGGACGCAG
* MKI67 *	AATTTGCTTGGAAAACAGTTTCA	TGCACTGAAGAACACATTTCCT
* NAMPT *	CAAGTTGCTGCCACCTTATCT	TGTTTCATGCCTTCTACAATCTCT
* SIRT1 *	GCCTCACATGCAAGCTCTAGT	TGTTCGAGGATCTGTGCCAA
* SLC5A1 *	CTGGCAGGCCGAAGTATG	CCACTTCCAATGTTACTAGCAAAG
* SLC5A2 *	GTTGCTGGATTCGAGTGGA	AGGTACACGGGTGCAAACA
* SNAI1 *	GGATCTCCAGGCTCGAAAGG	TGGCTTCGGATGTGCATCTT
* TJP1 *	ACACTGCTGAGTCCTTTGGT	ATCACAGTGTGGTAAGCGCA
* VIM *	TGGTCTAACGGTTTCCCCTA	GACCTCGGAGCGAGAGTG
* ZEB1 *	AGCTGTTTCAAGATGTTTCCTTCC	ACGAAAGCAGTGATTTTAATGATGG

## Data Availability

The original contributions presented in this study are included in the article. Further inquiries can be directed to the corresponding author.

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
