# Peer review of "Dual Roles of Canagliflozin on Cholangiocarcinoma Cell Growth and Enhanced Growth Suppression in Combination with FK866"

_ijms, 2025, doi:10.3390/ijms26030978_

Round 1
Reviewer 1 Report
Comments and Suggestions for Authors
The manuscript describes tumor-cell promoting effects of CANA at low concentrations - as are relevant for pharmacologic usage - and inhibitory effects at high doses. For my understanding the methods are apt, the results clearly described.
Throughout the manuscript abbreviations are used without explanation; for the non-expert reader it should be clarified at first mention what these abbreviations indicate, for others a quick reference always is useful.
For a transfer to humans I am not aware of any epidemiological study investigating tumor incidences in treated patients; transfer of animal studies is tricky, from in vitro data nearly impossible as mentioned in the manuscript.
As a minor point in lines 38ff: „demonstrated favorable anti-tumor effects in vitro and in vivo in several carcinomas, including liver, pancreatic, breast, thyroid, stomach, and lung cancers [9-17]“ the phrase implies in vivo results, i.e. reduced number of tumors at least in animal experiments. The cited sources do not support this claim, it should be phrased more carefully (possibly „and improvement in some tumor-associated parameters in animal experiments“).
A few misprints (capitalization) should be erased.
Author Response
Dear Editors and Reviewers,
Thank you very much for giving us a chance to revise our manuscript. We appreciate the comments, which have helped us to greatly improve the report. We have revised the manuscript as per the Reviewers’ comments and the revisions were marked by red pen. Now we try to answer the questions and comments one by one.
Some reviewers recommended that authors should perform western blotting, knock-down or overexpression for several genes, invasion assay, and animal and xenograft studies, but, unfortunately, we were told to revise within 10 days. We are afraid that we do not have enough time to respond and perform all of experiments suggested by reviewers. Authors may need to discuss it with the Editors.
Reviewer 1
> Throughout the manuscript abbreviations are used without explanation; for the non-expert reader it should be clarified at first mention what these abbreviations indicate, for others a quick reference always is useful.
Thank you very much for this comment. According to the suggestion, we have added abbreviation section (Lines 455-).
(Lines 455-)
Abbreviations
CANA canagliflozin
CCA cholangiocarcinoma
EMT epithelial-mesenchymal transition
HCC hepatocellular carcinoma
HDAC6 histone deacetylase 6
NAMPT nicotinamide phosphoribosyltransferase
SGLT sodium glucose cotransporter
SIRT1 sirtuin 1
TCGA the Cancer Genome Atlas
> For a transfer to humans I am not aware of any epidemiological study investigating tumor incidences in treated patients; transfer of animal studies is tricky, from in vitro data nearly impossible as mentioned in the manuscript.
Authors appreciate this important comment. Although in xenograft models of other types of cancers, SGLT2 inhibitors could inhibit the growth of tumors in animal models (PNAS. 2015;112:E4111-9, Int J Cancer. 2018;142:1712-22). Among those studies, the concentrations of the drug used in vitro study were similar to those in our present study. In addition, there are several epidemiological studies investigating the incidence of malignancies in the patients treated with SGLT2 inhibitors, showing that the incidence was lower in the treated group (J Gastroenterol. 2024;59:1120-32, J Natl Compr Canc Netw. 2024;22:e237118, Diabetes Metab Res Rev. 2024;40:e3784). From the results of above studies, authors guessed there may be possibility that the drug can show anti-tumor effects for cholangiocarcinoma in both animal and human studies. The manuscript has been revised (Lines 295-).
(Lines 295-) Our study had several limitations. First, CANA could only be studied in vitro, and the IC50 concentrations of CANA obtained from the in vitro studies were 5–15-fold higher than the actual blood concentrations after regular doses of CANA intake in humans. However, CANA has been extensively investigated in other cancer cells in xenograft and in vivo carcinogenesis models, where the CANA concentrations used were 50–100 times higher than those used in clinical settings. The dose of CANA used in these studies consistently showed good anti-tumor effects with fewer adverse events such as hypoglycemia and weight loss [10, 11, 15]. In addition, there is a report of anti-tumor effects even at lower concentrations of CANA than the IC50 calculated in vitro for liver cancer if administered for a prolonged period of time [10], suggesting that the serum concentration may not be needed to reach the level required in vitro because the drug is administered for a longer period of time. Moreover, there are several epidemiological studies investigating the incidence of malignancies in the patients treated with SGLT2 inhibitors, showing that the incidence was lower in the treated group [36-38]. From the results of these studies, we speculate there is possibility that SGLT2 inhibitors can show anti-tumor effects for CCA in both animal and human studies. The tumor-promoting effect of CANA at low concentrations observed in our study may hinder its clinical application in cancer chemotherapy; however, this may be overcome by agents that inhibit the NAD+ salvage pathway, such as NAMPT inhibitors. Based on the results of the present study, animal CCA models or patient-derived xenograft studies would provide further validation of the dual effects of CANA. Second, this study did not investigate protein levels but mainly gene expressions of major molecules. In addition, a multifaceted evaluation was not conducted for apoptosis and migration/invasion. Therefore, more detailed analysis of protein expression, apoptosis, and migration/invasion will provide a more comprehensive understanding for the effects of CANA against malignancy.
> As a minor point in lines 38ff: „demonstrated favorable anti-tumor effects in vitro and in vivo in several carcinomas, including liver, pancreatic, breast, thyroid, stomach, and lung cancers [9-17]“ the phrase implies in vivo results, i.e. reduced number of tumors at least in animal experiments. The cited sources do not support this claim, it should be phrased more carefully (possibly „and improvement in some tumor-associated parameters in animal experiments“).
Thank you for this insightful comment. We have revised the manuscript (Lines 38-).
(Lines 38-) The sodium-glucose cotransporter 2 (SGLT2) inhibitors, including canagliflozin (CANA), have been developed for type 2 diabetes treatment by preventing the reabsorption of glucose and increasing its urinary excretion [9]. CANA has demonstrated favorable anti-tumor effects in vitro, growth suppression effects on xenografts, improvement effects on some tumor-associated parameters in animal experiments for various types of cancers, including liver, pancreatic, breast, thyroid, stomach, and lung cancers [10-18].
> A few misprints (capitalization) should be erased.
We are afraid and unfortunately not able to figure out which are misprints (capitalization) needed to be erased. If they are so, editorial staff will correct them for publication.
Reviewer 2 Report
Comments and Suggestions for Authors
Title: Dual roles of canagliflozin on cholangiocarcinoma cell growth and enhanced growth suppression in combination with FK866.
Thanks to the authors for the interesting manuscript.
My comments:
1. Huh7 is a good control for SLC5A1, but did authors look at Panc-1 or Panc-2 cell line for example? These cells too have higher SLC5A1 expression.
2. Why was gene expression of SLC5A1 higher in HuCCT1 cells?
3. Can a patient with CCA receive CANA if they have type 1 diabetes?
4. If CANA has potential tumor growing effect why would authors recommend for as anti-cancer agent?
5. Did authors observe issues with the apoptosis assay? I know sometimes the assay used can develop problems especially during trypsinization of the cells.
6. Were authors able to account for the toxicity produced by the MTS solution and why was the cell density 2.5x103?
7. Can you be specific in your introduction that CANA is used for type-2 diabetes?
Author Response
Dear Editors and Reviewers,
Thank you very much for giving us a chance to revise our manuscript. We appreciate the comments, which have helped us to greatly improve the report. We have revised the manuscript as per the Reviewers’ comments and the revisions were marked by red pen. Now we try to answer the questions and comments one by one.
Some reviewers recommended that authors should perform western blotting, knock-down or overexpression for several genes, invasion assay, and animal and xenograft studies, but, unfortunately, we were told to revise within 10 days. We are afraid that we do not have enough time to respond and perform all of experiments suggested by reviewers. Authors may need to discuss it with the Editors.
Reviewer 2
> 1. Huh7 is a good control for SLC5A1, but did authors look at Panc-1 or Panc-2 cell line for example? These cells too have higher SLC5A1 expression.
Thank you for your pointing. We did not check SLC5A1 level in other cell lines, but we just referred to a previous paper in which the levels of SLC5A1 (SGLT2) were higher among liver cancer cells and this cell line was chosen as a control (Int J Cancer. 2018;142:1712-22, doi:10.1002/ijc.31193). We have revised the manuscript (Lines 63-).
(Line 63-) The expression of genes encoding sodium glucose cotransporter 1 (SGLT1) and SGLT2 in each CCA cell line was evaluated and compared with that in previously re-ported hepatocellular carcinoma (HCC) cell lines [10]. According to the report, the HCC cell line HLE was used as a control for low expression of SLC5A1 and SLC5A2, and Huh7 was used as a control for high expression of SLC5A1 and as a reference for SLC5A2 expression in this study. Gene expression of SLC5A1 was significantly higher in HuCCT1 cells than in HLE cells and significantly lower in Huh28 cells than in Huh7 cells (Fig 1A). The gene expression of SLC5A2 was significantly lower in TFK-1 cells than in Huh7 cells and significantly higher in Huh28 cells than in HLE cells (Fig 1B).
> 2. Why was gene expression of SLC5A1 higher in HuCCT1 cells?
Thank you for this interesting comment, but we are not sure. Other types of cancer, for example liver cancer, also show various expression levels of SLC5A1 from higher to similar compared to normal cells (Int J Cancer. 2018;142:1712-22, doi:10.1002/ijc.31193). It may be due to their origin, degree of differentiation, or race. Human tissue samples of CCA also show varied expression according to the data from TCGA (Figure 1), but the reason and mechanism have not been examined in detail.
> 3. Can a patient with CCA receive CANA if they have type 1 diabetes?
A meta-analysis has shown a significant benefit of SGLT2 inhibitors for individuals with the efficacy type 1 diabetes (Metabolism. 2024:153:155791. doi: 10.1016/j.metabol.2024.155791). There are many types of SGLT2 inhibitors used in clinical practice, including CANA. Among them, some can be used for type 1 diabetes as well as type 2, but it seems to depend on the health insurance system in each country. For example, in Japan, dapagliflozin and ipragliflozin can be prescribed for type 1 diabetes. If a patient with CCA needs an SGLT2 inhibitor because of type 1 diabetes, the medicine, including CANA, may be able to be prescribed without severe adverse events.
> 4. If CANA has potential tumor growing effect why would authors recommend for as anti-cancer agent?
Thank you for this fundamental comment. Basically, we do not fully recommend this medicine for anti-cancer agent, but we do suggest the possibility to be effective with care. There have been many scientific papers in which SGLT2 inhibitors possess anti-cancer effects of and the authors recommend them as anti-cancer agents, but in this study we find they may have opposite effects and we suggest they should be used with caution. We have revised the manuscript (Lines 330-).
(Lines 330-) In conclusion, CANA inhibited tumor growth and EMT by inducing cell cycle arrest in CCA cells, and its anti-tumor effect was enhanced when combined with NAMPT inhibitors, while it may also exert tumor-promoting effects via the NAD+ salvage pathway and SIRT1 activation. Further studies are warranted to investigate the dual effects of CANA in CCA. Although much attention has been focused solely on anti-tumor effects of SGLT2 inhibitors, including CANA, we suggest that the opposite effect should also be noted.
> 5. Did authors observe issues with the apoptosis assay? I know sometimes the assay used can develop problems especially during trypsinization of the cells.
Thank you for your comment. As this reviewer points out, trypsinization can develop problem for apoptosis assay; therefore, according to the protocol (BD Pharmingen FITC Annexin V Apoptosis Detection Kit I), we performed the experiment properly.
> 6. Were authors able to account for the toxicity produced by the MTS solution and why was the cell density 2.5x103?
The MTS solution was used at the same concentration and volume for all samples including controls; therefore, even if MTS itself has cytotoxic, a relative evaluation is possible. Because the cell line HuCCT1 proliferates relatively rapidly, we seeded at a lower concentration of 2.5x10^3, although 5.0x10^3-1.0x10^4 cells are generally seeded for 96 wells.
> 7. Can you be specific in your introduction that CANA is used for type-2 diabetes?
Authors appreciate this important comment. We have revised the manuscript (Lines 38-).
(Lines 38-) The sodium-glucose cotransporter 2 (SGLT2) inhibitors, including canagliflozin (CANA), have been developed for type 2 diabetes treatment by preventing the reabsorption of glucose and increasing its urinary excretion [9]. CANA has demonstrated favorable anti-tumor effects in vitro, growth suppression effects on xenografts, improvement effects on some tumor-associated parameters in animal experiments for various types of cancers, including liver, pancreatic, breast, thyroid, stomach, and lung cancers [10-18].
Reviewer 3 Report
Comments and Suggestions for Authors
Comments to the authors:
I have some concerns that need to be addressed by the authors:
1: The concentrations of CANA used (30–80 µM) are much higher than clinically achievable plasma levels in patients. This raises doubts about the translational relevance of the findings. The authors should include justification for using such high doses and discuss how these concentrations align with in vivo pharmacokinetics.
2: The findings are based solely on in vitro data, which limits clinical applicability. An animal model or patient-derived xenograft (PDX) study would strengthen the conclusion and provide further validation of CANA's dual effects.
3: While CANA’s effects on NAD+/SIRT1 and HDAC6 pathways are suggested, the mechanistic link between these effects and tumor growth is not fully elucidated. The authors should p erform additional experiments (e.g., Western blot) to confirm changes in protein levels of SIRT1, HDAC6, and key cell cycle regulators. Knockdown or overexpression experiments for SIRT1/NAMPT would clarify their roles in the observed dual effects.
4: The paper reports increased cell viability and S-phase progression at low concentrations of CANA, but the underlying mechanism remains unclear. Investigate whether low-dose CANA induces metabolic changes or compensatory upregulation of other pathways (e.g., glycolysis) and explore the role of glucose transporter activity in this phenomenon.
5: The limited effect on apoptosis contrasts with findings from previous studies in other cancer types (e.g., liver and thyroid cancers). This discrepancy is not fully addressed. The authors should perform further apoptosis-specific assays (e.g., caspase activation, TUNEL assay) at higher concentrations and discuss potential differences in cancer cell metabolism or resistance mechanisms.
6: The comparison of SGLT1/SGLT2 gene expression using TCGA data is informative but lacks validation in clinical CCA samples. Also Include clinical validation (e.g., immunohistochemistry or RNA sequencing) of SGLT1/SGLT2 expression in patient tissues.
7: The paper concludes that CANA inhibits EMT, but some mesenchymal markers (ZEB1, SNAI1) were paradoxically upregulated at high doses. Explore the functional significance of increased mesenchymal marker expression. Include additional assays (e.g., invasion assays) to further validate EMT inhibition.
Minor Comments:
1: Ensure consistent terminology for gene and protein names (e.g., SIRT1/Sirt1).
2: Reformat figures to improve readability (e.g., clearer legends, larger axis labels).
3: Include additional references where discrepancies with previous studies are discussed.
Author Response
Dear Editors and Reviewers,
Thank you very much for giving us a chance to revise our manuscript. We appreciate the comments, which have helped us to greatly improve the report. We have revised the manuscript as per the Reviewers’ comments and the revisions were marked by red pen. Now we try to answer the questions and comments one by one.
Some reviewers recommended that authors should perform western blotting, knock-down or overexpression for several genes, invasion assay, and animal and xenograft studies, but, unfortunately, we were told to revise within 10 days. We are afraid that we do not have enough time to respond and perform all of experiments suggested by reviewers. Authors may need to discuss it with the Editors.
Reviewer 3
> 1: The concentrations of CANA used (30–80 µM) are much higher than clinically achievable plasma levels in patients. This raises doubts about the translational relevance of the findings. The authors should include justification for using such high doses and discuss how these concentrations align with in vivo pharmacokinetics.
We appreciate this important comment. As pointed, the serum Cmax of CANA is reported as 7.8 µM when administered 300 mg/day and this is lower than that used in in vitro experiment. Although only in vitro studies were conducted in this study, animal studies of pancreatic cancer have demonstrated tumor suppression at much lower concentrations of CANA than the IC50 calculated in vitro (Int J Oncol. 2020;57:1223-33). In addition, there is a report of anti-tumor effects even at low concentrations of CANA for liver cancer if administered for a prolonged period of time (Int J Cancer. 2018;142:1712-22). These studies suggest that the concentration may not be needed to reach the level required in vitro because the drug is administered for a longer period of time. Moreover, there are several epidemiological studies investigating the incidence of malignancies in the patients treated with SGLT2 inhibitors, showing that the incidence was lower in the treated group (J Gastroenterol. 2024;59:1120-32, J Natl Compr Canc Netw. 2024;22:e237118, Diabetes Metab Res Rev. 2024;40:e3784). Therefore, even clinically achievable plasma levels of CANA can show suppressive effects against malignancy. The manuscript has been revised (Lines 295-).
(Lines 295-) Our study had several limitations. First, CANA could only be studied in vitro, and the IC50 concentrations of CANA obtained from the in vitro studies were 5–15-fold higher than the actual blood concentrations after regular doses of CANA intake in humans. However, CANA has been extensively investigated in other cancer cells in xenograft and in vivo carcinogenesis models, where the CANA concentrations used were 50–100 times higher than those used in clinical settings. The dose of CANA used in these studies consistently showed good anti-tumor effects with fewer adverse events such as hypoglycemia and weight loss [10, 11, 15]. In addition, there is a report of anti-tumor effects even at lower concentrations of CANA than the IC50 calculated in vitro for liver cancer if administered for a prolonged period of time [10], suggesting that the serum concentration may not be needed to reach the level required in vitro because the drug is administered for a longer period of time. Moreover, there are several epidemiological studies investigating the incidence of malignancies in the patients treated with SGLT2 inhibitors, showing that the incidence was lower in the treated group [36-38]. From the results of these studies, we speculate there is possibility that SGLT2 inhibitors can show anti-tumor effects for CCA in both animal and human studies. The tumor-promoting effect of CANA at low concentrations observed in our study may hinder its clinical application in cancer chemotherapy; however, this may be overcome by agents that inhibit the NAD+ salvage pathway, such as NAMPT inhibitors. Based on the results of the present study, animal CCA models or patient-derived xenograft studies would provide further validation of the dual effects of CANA. Second, this study did not investigate protein levels but mainly gene expressions of major molecules. In addition, a multifaceted evaluation was not conducted for apoptosis and migration/invasion. Therefore, more detailed analysis of protein expression, apoptosis, and migration/invasion will provide a more comprehensive understanding for the effects of CANA against malignancy.
> 2: The findings are based solely on in vitro data, which limits clinical applicability. An animal model or patient-derived xenograft (PDX) study would strengthen the conclusion and provide further validation of CANA's dual effects.
We thank for this suggestive comment. Based on the results of this study, animal CCA models or patient-derived xenograft studies would validate CANA's dual effects. However, unfortunately, we were told to revise within 10 days. We are afraid that we do not have enough time to respond and perform all of experiments suggested by reviewers. Authors may need to discuss it with the Editors. The manuscript has been revised (Lines 308-).
(Lines 308-) From the results of these studies, we speculate there is possibility that SGLT2 inhibitors can show anti-tumor effects for CCA in both animal and human studies. The tumor-promoting effect of CANA at low concentrations observed in our study may hinder its clinical application in cancer chemotherapy; however, this may be overcome by agents that inhibit the NAD+ salvage pathway, such as NAMPT inhibitors. Based on the results of the present study, animal CCA models or patient-derived xenograft studies would provide further validation of the dual effects of CANA. Second, this study did not investigate protein levels but mainly gene expressions of major molecules. In addition, a multifaceted evaluation was not conducted for apoptosis and migration/invasion. Therefore, more detailed analysis of protein expression, apoptosis, and migration/invasion will provide a more comprehensive understanding for the effects of CANA against malignancy.
> 3: While CANA’s effects on NAD+/SIRT1 and HDAC6 pathways are suggested, the mechanistic link between these effects and tumor growth is not fully elucidated. The authors should perform additional experiments (e.g., Western blot) to confirm changes in protein levels of SIRT1, HDAC6, and key cell cycle regulators. Knockdown or overexpression experiments for SIRT1/NAMPT would clarify their roles in the observed dual effects.
Authors thank for these important comments. As pointed, to clarify the roles of key molecules in the observed dual effects of CANA on CCA, western blot, knock-down or overexpression experiments would be necessary. However, unfortunately, as described above we were told to revise within 10 days. We are afraid that we do not have enough time to respond and perform all of experiments suggested by reviewers, including this reviewer and others. Authors may need to discuss it with the Editors. The manuscript has been revised (Lines 308-).
(Lines 308-) From the results of these studies, we speculate there is possibility that SGLT2 inhibitors can show anti-tumor effects for CCA in both animal and human studies. The tumor-promoting effect of CANA at low concentrations observed in our study may hinder its clinical application in cancer chemotherapy; however, this may be overcome by agents that inhibit the NAD+ salvage pathway, such as NAMPT inhibitors. Based on the results of the present study, animal CCA models or patient-derived xenograft studies would provide further validation of the dual effects of CANA. Second, this study did not investigate protein levels but mainly gene expressions of major molecules. In addition, a multifaceted evaluation was not conducted for apoptosis and migration/invasion. Therefore, more detailed analysis of protein expression, apoptosis, and migration/invasion will provide a more comprehensive understanding for the effects of CANA against malignancy. Third, CANA treatment-induced metabolic changes and compensatory alterations of other pathways, such as glucose transporter type 1 upregulation or glycolysis, were not investigated in this study. A previous study demonstrated glucose uptake in cancer cells was decreased by CANA in a dose-dependent manner [10], suggesting that the cancer cells, at least in part, depend on SGLT2 for glucose uptake and compensate mechanisms for glucose uptake may not work after CANA treatment. In the present study, SGLT2 gene expression was upregulated by the treatment with CANA. This upregulation might be a compensate mechanism of CCA cells, although the SGLT2 activity and glucose uptake have not been examined. Further research is required in order to reveal how cell viability increased and S-phase progressed by low dose treatment of CANA.
> 4: The paper reports increased cell viability and S-phase progression at low concentrations of CANA, but the underlying mechanism remains unclear. Investigate whether low-dose CANA induces metabolic changes or compensatory upregulation of other pathways (e.g., glycolysis) and explore the role of glucose transporter activity in this phenomenon.
We appreciate these suggestive comments. A previous study using liver cancer cells (Int J Cancer. 2018;142:1712-22) demonstrated glucose uptake in the cells was decreased by CANA in a concentration-dependent manner. This result may suggest that the cancer cells, at least in part, depend on SGLT2 for glucose uptake and compensate mechanisms for glucose uptake do not work after CANA treatment. In the present study, SGLT2 gene expression was upregulated by the treatment with CANA. This upregulation might be a compensate mechanism of CCA cells, although the actual SGLT2 activity and glucose uptake have not been examined. With regard to underlying mechanism of increased cell viability and S-phase progression at low concentrations of CANA, we speculated that activated NAD+ salvage pathway played an important role. As pointed, metabolic changes or compensatory upregulation of other pathways, such as GLUT1 upregulation and glycolysis, might be important for increased cell viability and S-phase progression by CANA treatment; therefore, further investigation would figure out the role of glucose transporter activity in CCA cells. However, unfortunately, as described above we were told to revise within 10 days. Again, we are afraid that we have not enough time to respond and perform all of experiments suggested by reviewers, including this reviewer and others. Authors may need to discuss it with the Editors. The manuscript has been revised (Lines 308-).
(Lines 308-) From the results of these studies, we speculate there is possibility that SGLT2 inhibitors can show anti-tumor effects for CCA in both animal and human studies. The tumor-promoting effect of CANA at low concentrations observed in our study may hinder its clinical application in cancer chemotherapy; however, this may be overcome by agents that inhibit the NAD+ salvage pathway, such as NAMPT inhibitors. Based on the results of the present study, animal CCA models or patient-derived xenograft studies would provide further validation of the dual effects of CANA. Second, this study did not investigate protein levels but mainly gene expressions of major molecules. In addition, a multifaceted evaluation was not conducted for apoptosis and migration/invasion. Therefore, more detailed analysis of protein expression, apoptosis, and migration/invasion will provide a more comprehensive understanding for the effects of CANA against malignancy. Third, CANA treatment-induced metabolic changes and compensatory alterations of other pathways, such as glucose transporter type 1 upregulation or glycolysis, were not investigated in this study. A previous study demonstrated glucose uptake in cancer cells was decreased by CANA in a dose-dependent manner [10], suggesting that the cancer cells, at least in part, depend on SGLT2 for glucose uptake and compensate mechanisms for glucose uptake may not work after CANA treatment. In the present study, SGLT2 gene expression was upregulated by the treatment with CANA. This upregulation might be a compensate mechanism of CCA cells, although the SGLT2 activity and glucose uptake have not been examined. Further research is required in order to reveal how cell viability increased and S-phase progressed by low dose treatment of CANA.
> 5: The limited effect on apoptosis contrasts with findings from previous studies in other cancer types (e.g., liver and thyroid cancers). This discrepancy is not fully addressed. The authors should perform further apoptosis-specific assays (e.g., caspase activation, TUNEL assay) at higher concentrations and discuss potential differences in cancer cell metabolism or resistance mechanisms.
Thank you very much for this fundamental comment. As pointed, detailed analyses for evaluating apoptosis by CANA treatment. However, unfortunately, as described above we were told to revise within 10 days. Again, we are afraid that we do not have enough time to respond and perform all of experiments suggested by reviewers, including this reviewer and others. Authors may need to discuss it with the Editors whether all of those should be done for publication. The manuscript has been revised (Lines 308-).
(Lines 308-) From the results of these studies, we speculate there is possibility that SGLT2 inhibitors can show anti-tumor effects for CCA in both animal and human studies. The tumor-promoting effect of CANA at low concentrations observed in our study may hinder its clinical application in cancer chemotherapy; however, this may be overcome by agents that inhibit the NAD+ salvage pathway, such as NAMPT inhibitors. Based on the results of the present study, animal CCA models or patient-derived xenograft studies would provide further validation of the dual effects of CANA. Second, this study did not investigate protein levels but mainly gene expressions of major molecules. In addition, a multifaceted evaluation was not conducted for apoptosis and migration/invasion. Therefore, more detailed analysis of protein expression, apoptosis, and migration/invasion will provide a more comprehensive understanding for the effects of CANA against malignancy.
> 6: The comparison of SGLT1/SGLT2 gene expression using TCGA data is informative but lacks validation in clinical CCA samples. Also Include clinical validation (e.g., immunohistochemistry or RNA sequencing) of SGLT1/SGLT2 expression in patient tissues.
Thank you for this comment. Validation using clinical sample must be important and we would like to do. However, as described above we were told to revise within 10 days. Again, we are afraid that we do not have enough time to respond and perform all of experiments suggested by reviewers, including this reviewer and others. Authors may need to discuss it with the Editors whether all of those should be done for publication.
> 7: The paper concludes that CANA inhibits EMT, but some mesenchymal markers (ZEB1, SNAI1) were paradoxically upregulated at high doses. Explore the functional significance of increased mesenchymal marker expression. Include additional assays (e.g., invasion assays) to further validate EMT inhibition.
We thank for this suggestive comment. For further validation of inhibiting EMT, other assay, including invasion assays are needed. However, unfortunately, as described above we were told to revise within 10 days. Again, we are afraid that we do not have enough time to respond and perform all of experiments suggested by reviewers, including this reviewer and others. Authors may need to discuss it with the Editors whether all of those should be done for paper acceptance. We have revised the manuscript (Lines 308-).
(Lines 308-) From the results of these studies, we speculate there is possibility that SGLT2 inhibitors can show anti-tumor effects for CCA in both animal and human studies. The tumor-promoting effect of CANA at low concentrations observed in our study may hinder its clinical application in cancer chemotherapy; however, this may be overcome by agents that inhibit the NAD+ salvage pathway, such as NAMPT inhibitors. Based on the results of the present study, animal CCA models or patient-derived xenograft studies would provide further validation of the dual effects of CANA. Second, this study did not investigate protein levels but mainly gene expressions of major molecules. In addition, a multifaceted evaluation was not conducted for apoptosis and migration/invasion. Therefore, more detailed analysis of protein expression, apoptosis, and migration/invasion will provide a more comprehensive understanding for the effects of CANA against malignancy.
Minor Comments:
> 1: Ensure consistent terminology for gene and protein names (e.g., SIRT1/Sirt1).
We understand human genes are written in uppercase and describe them in uppercase italic letters in figures. Editorial staff will instruct whether they should be italic or not in the text.
> 2: Reformat figures to improve readability (e.g., clearer legends, larger axis labels).
The figures have been reformatted according to the recommendation by this and other reviewers.
> 3: Include additional references where discrepancies with previous studies are discussed.
Several references have been added for discussion.
Reviewer 4 Report
Comments and Suggestions for Authors
The study addresses an important topic and demonstrates scientific rigor, but several critical areas require further attention to improve its clarity, transparency, and impact.
The introduction should emphasize the novelty of this study, especially regarding CANA’s dual role and its interactions with SIRT1 and the NAD+ pathway. The authors should clearly articulate how their study builds on or diverges from prior research, particularly with respect to mechanistic insights or therapeutic implications.
Methods: The rationale for selecting specific CANA concentrations is unclear. Since the IC50 varies across different cell lines, it would be useful to justify the choice of concentration and explain how these concentrations relate to potential clinical relevance. Further clarification is also needed on how "clinically relevant biomarkers" were defined and identified. This would help improve the transparency and reproducibility of the study.
Line 75-82: Include confidence intervals for IC50 values for each cell line to provide a clearer understanding of variability and experimental robustness.
Line 120-125: The apoptosis assay results suggest limited effects at 50 µM. It would be useful to note whether higher concentrations were tested and, if so, why they were not included in the main analysis.
Line 200-220: Annotate Figures 2 and 3 with statistical indicators such as p-values and confidence intervals. This will allow readers to interpret the significance of the observed differences easily.
Line 265-275: The statement about the NAD+ salvage pathway promoting tumor growth could be further developed. Was this finding anticipated or unexpected? Discuss the implications of this observation, especially in the context of cancer metabolism and possible therapeutic targeting of the pathway.
Line 280-300: Avoid overstating clinical applicability. The conclusions imply potential clinical translation, but this claim is premature without in vivo validation. A more balanced discussion is recommended.
Line 310-320: Address the limitations of using in vitro models. Highlight how the experimental concentrations of CANA used in vitro compared to what might be achievable or safe in vivo. This would provide a more complete view of the study’s translational relevance.
Figures 2 and 3: These figures should include more descriptive annotations, particularly p-values and confidence intervals, to clarify the findings' significance. Consider using color or other visual elements to highlight key differences in the data.
Table 3: Add context on the identified biomarkers and explain their potential relevance for cancer diagnostics or therapeutics. Discuss how these biomarkers might be used in future clinical studies or guide treatment strategies.
Comments on the Quality of English LanguageThe quality of English in the manuscript is sufficient to convey the primary findings. However, several areas could benefit from improvement for clarity and readability. Issues such as awkward phrasing, inconsistent terminology, and grammatical errors can detract from the overall presentation. For example, terms like "synergy" and "combination index" are used inconsistently, potentially confusing readers. Additionally, some sentences, particularly in the introduction and discussion, are overly complex and could be simplified. Professional language editing is recommended to enhance the manuscript's coherence, scientific precision, and overall readability.
Author Response
Dear Editors and Reviewers,
Thank you very much for giving us a chance to revise our manuscript. We appreciate the comments, which have helped us to greatly improve the report. We have revised the manuscript as per the Reviewers’ comments and the revisions were marked by red pen. Now we try to answer the questions and comments one by one.
Some reviewers recommended that authors should perform western blotting, knock-down or overexpression for several genes, invasion assay, and animal and xenograft studies, but, unfortunately, we were told to revise within 10 days. We are afraid that we do not have enough time to respond and perform all of experiments suggested by reviewers. Authors may need to discuss it with the Editors.
Reviewer 4
The study addresses an important topic and demonstrates scientific rigor, but several critical areas require further attention to improve its clarity, transparency, and impact.
> The introduction should emphasize the novelty of this study, especially regarding CANA’s dual role and its interactions with SIRT1 and the NAD+ pathway. The authors should clearly articulate how their study builds on or diverges from prior research, particularly with respect to mechanistic insights or therapeutic implications.
We thank for this suggestive comment. The manuscript has been revised (Lines 44-).
(Lines 44-) The anti-tumor mechanism of CANA remains unclear, as multiple mechanisms other than SGLT2 inhibition have been speculated [13, 17, 18]. CANA reduces oxidative stress and improves energy metabolism through the activation of sirtuin 1 (SIRT1), an NAD+-dependent class III histone deacetylase, in some non-tumor tissues [19-21], and the activation of SIRT1 has been associated with increased healthy life expectancy [22]. In contrast, SIRT1 activation promotes tumor growth and epithelial-mesenchymal transition (EMT) [23-25]. However, studies analyzing the efficacy of CANA in CCA or the effect of SIRT1 activation by CANA on cancer cells are lacking.
In the present study, the anti-cancer properties of CANA against human CCA cells are reported. We demonstrate that CANA significantly inhibits cell growth of CCA with both higher and lower expression of SGLT2 by arresting cell cycle. This study also indicates that CANA inhibits cancer cell migration, which is a firstly re-ported mechanism. Interestingly, CANA exerts tumor-promoting effects through the NAD+ salvage pathway and SIRT1 activation when treated with lower concentration, suggesting that CANA may have dual roles on CCA cell growth. While anti-cancer effects of SGLT2 inhibitors have been mainly focused, we suggest that there may also be opposite effect on malignancy.
> Methods: The rationale for selecting specific CANA concentrations is unclear. Since the IC50 varies across different cell lines, it would be useful to justify the choice of concentration and explain how these concentrations relate to potential clinical relevance. Further clarification is also needed on how "clinically relevant biomarkers" were defined and identified. This would help improve the transparency and reproducibility of the study.
Thank you for this important comment. As pointed, IC50 varies among different cell lines. In general, concentrations of an agent used for in vitro study are determined by using a wide range at first or by referring to previous reports. Also, treatment concentrations in in vitro experiments are often higher than those of blood concentrations of the agent used in clinical practice. In fact, a previous study (Int J Cancer. 2018;142:1712-22, doi:10.1002/ijc.31193) showed that the concentrations of CANA used for in vitro study were much higher compared to serum levels reported in clinical trial (less than 10 μM). In the present study, we referred to the above study and treated CCA cells with similar concentrations of CANA, then performed cell proliferation assay, and calculated and determined IC50 for each cell. We found a novel phenomenon that cell proliferation was promoted by relatively low dose (30 μM) of CANA treatment in HuCCT1. Therefore, the concentrations of CANA for HuCCT1 were set as follows: 30 µM, 50 µM, which is IC50, and 80 µM, which is the maximum concentration that can be adjusted with considering the effect of the solvent DMSO.
As for “potential clinical relevance,” CANA has been extensively investigated in other cancer cells in xenograft and in vivo carcinogenesis models, where the CANA concentrations used were 50–100 times higher than those used in clinical settings. The dose of CANA used in these studies consistently showed good anti-tumor effects with fewer adverse events such as hypoglycemia and weight loss (Int J Cancer 2018;142:1712-22, Int J Oncol. 2020;57:1223-33, Cancer Cell Int. 2022; 22:74). In addition, there is a report of anti-tumor effects even at lower concentrations of CANA than the IC50 calculated in vitro for liver cancer if administered for a prolonged period of time (Int J Cancer. 2018;142:1712-22), suggesting that the serum concentration may not be needed to reach the level required in vitro because the drug is administered for a longer period of time. Moreover, there are several epidemiological studies investigating the incidence of malignancies in the patients treated with SGLT2 inhibitors, showing that the incidence was lower in the treated group (J Gastroenterol. 2024;59:1120-32, J Natl Compr Canc Netw. 2024;22:e237118, Diabetes Metab Res Rev. 2024;40:e3784). From the results of these studies, we speculate there is possibility that SGLT2 inhibitors can show anti-tumor effects for CCA in both animal and human studies.
Unfortunately, authors are not sure what this reviewer points out about “clinically relevant biomarkers.” If this refers to the expression of SLGT2 in cancer tissues, clinicians can analyze the obtained CCA tissue samples to examine its expression as a companion diagnosis, which would recommend SGLT2 inhibitors to be used in the clinical practice for CCA treatment. If this reviewer means biomarkers in blood, there has been no report for molecules in blood associated with tissue SGLT2 expression.
The manuscript has been revised according to the comments (Line 221-, 295-).
(Lines 221-) In addition, additional anti-tumor effects of CANA have been reported in SGLT2 knockdown cells [13]. CANA binds specifically to HDAC6 and directly inhibits HDAC6 [18]. Furthermore, HDAC6 inhibition reduces CCA cell growth by restoring primary cilia [27]. These reports support our results that CANA reduces HDAC6 gene expression in CCA in a concentration-dependent manner and inhibits tumor growth. These findings demonstrate that, in addition to sodium glucose cotransporters (SGLTs) inhibition, multiple mechanisms are involved in the anti-tumor effect of CANA.
Most of the papers to date examining the anti-cancer effects of SGLT2 inhibitors have been focused on SGLT2 expression in cancer cells. That is, the inhibitors are presumably effective only on SGLT2-expressing cancer cells [10, 28]. If so, the agent can be used in the future for selected patients with tumors expressing SGLT2, as a biomarker, based on the results of companion diagnosis. However, previous reports and our pre-sent study indicated that CANA has anti-tumor effects regardless of SGLT2 expression in the cells [13, 27]. Since the anti-cancer effects of SGLT2 inhibitors may depend on the type of cancer and/or the type of SGLT2 inhibitor, further research focusing on them may be able to clarify detailed mechanisms.
(Lines 295-) Our study had several limitations. First, CANA could only be studied in vitro, and the IC50 concentrations of CANA obtained from the in vitro studies were 5–15-fold higher than the actual blood concentrations after regular doses of CANA intake in humans. However, CANA has been extensively investigated in other cancer cells in xenograft and in vivo carcinogenesis models, where the CANA concentrations used were 50–100 times higher than those used in clinical settings. The dose of CANA used in these studies consistently showed good anti-tumor effects with fewer adverse events such as hypoglycemia and weight loss [10, 11, 15]. In addition, there is a report of anti-tumor effects even at lower concentrations of CANA than the IC50 calculated in vitro for liver cancer if administered for a prolonged period of time [10], suggesting that the serum concentration may not be needed to reach the level required in vitro because the drug is administered for a longer period of time. Moreover, there are several epidemiological studies investigating the incidence of malignancies in the patients treated with SGLT2 inhibitors, showing that the incidence was lower in the treated group [36-38]. From the results of these studies, we speculate there is possibility that SGLT2 inhibitors can show anti-tumor effects for CCA in both animal and human studies. The tumor-promoting effect of CANA at low concentrations observed in our study may hinder its clinical application in cancer chemotherapy; however, this may be overcome by agents that inhibit the NAD+ salvage pathway, such as NAMPT inhibitors. Based on the results of the present study, animal CCA models or patient-derived xenograft studies would provide further validation of the dual effects of CANA. Second, this study did not investigate protein levels but mainly gene expressions of major molecules. In addition, a multifaceted evaluation was not conducted for apoptosis and migration/invasion. Therefore, more detailed analysis of protein expression, apoptosis, and migration/invasion will provide a more comprehensive understanding for the effects of CANA against malignancy.
> Line 75-82: Include confidence intervals for IC50 values for each cell line to provide a clearer understanding of variability and experimental robustness.
Thank you for this comment. We have revised the manuscript (Lines 88-, 434-).
(Lines 88-) Cell viability assessment by 3-(4,5-dimethylthiazol-2-yl)-5-(3-carboxymethoxyphenyl)-2-(4-sulfophenyl)-2H-tetrazolium (MTS) assay showed that CANA induced a concentration-dependent decrease in survival of HuCCT1, Huh28, and TFK-1 with median inhibition concentration (IC50) and 95% confidence interval of 52.9 µM (50.6–55.2), 42.6 µM (40.2–45.1), and 46.1 µM (44.4–47.7), respectively. HuCCT1 showed a substantial increase in survival at 30 µM, and TFK-1 showed improved survival at lower doses (Fig 1D).
(Lines 434-) 4.9. Statistical analysis
All data are presented as the means ± standard error of the mean or mean ± confidence intervals. Unless other-wise stated, the experiments were conducted in parallel (n = 6 in each group). Non-parametric statistical analyses between three or more groups were performed using the Kruskal–Wallis test, followed by the Steel–Dwass test. A two-sided p value Ë‚ 0.05 indicated statistical significance. All statistical analyses were performed using the R version 4.3.1 software (The R Foundation for Statistical Computing, Vienna, Austria). The IC50 and two-sided 95% confidence interval were calculated using the drc package in the R software.
> Line 120-125: The apoptosis assay results suggest limited effects at 50 µM. It would be useful to note whether higher concentrations were tested and, if so, why they were not included in the main analysis.
Thank you for this important comment. The effects at higher concentration of CANA on apoptosis were also examined, but the rate of section for early apoptosis (Q4) was not changed. We have revised the manuscript (Lines 125-, 135-) and figure (Fig. 4).
(Lines 125-) Treatment with CANA did not affect the expression of BAX, which encodes the apoptotic regulator BAX, but significantly increased the expression of BCL2 and BCL2L1, which encode the apoptotic regulators Bcl-2 and Bcl-2-like protein 1, respectively, in a concentration-dependent manner (Fig 4A). To assess actual apoptosis, HuCCT1 was treated with 50 µM of CANA for 0–24 h for the apoptosis assay (Fig 4B). Compared with the controls, the percentage of cells in the Q2 region, which defines apoptosis and cell death, was unchanged after 24 h, and the percentage of cells in the Q4 region, which defines early apoptosis, showed a significant increase after 24 h; however, the increase was negligible (2%) (Fig 4C). At 80 µM of CANA for 24 h treatment, the percentage of cells in the Q4 region was not changed significantly (Fig 4D).
(Lines 136-) Figure 4. Effect of CANA on apoptosis in CCA cells. (A) The mRNA expressions of BAX, BCL2, and BCL2L1 in HuCCT1 cells after treatment with several graded concentrations of CANA were verified by qRT-PCR (n = 6). (B) Apoptosis of HuCCT1 cells at several time points after treatment with 50 µM CANA was verified by the Annexin V apoptosis assay. (C) The percentage of cells in the Q2 (late apoptotic cells and dead cells), Q3 (non-apoptotic cells), and Q4 (early apoptotic cells) regions was quantified. (n = 6). (D) Apoptosis of HuCCT1 cells treated with several concentrations of CANA for 24 h was verified by the Annexin V apoptosis assay. *p < 0.05 compared with the control group. Abbreviations: CANA, canagliflozin; CCA, cholangiocarcinoma; PI, propidium iodide; qRT-PCR, quantitative reverse transcription polymerase chain reaction.
> Line 200-220: Annotate Figures 2 and 3 with statistical indicators such as p-values and confidence intervals. This will allow readers to interpret the significance of the observed differences easily.
Thank you for this comment. We have revised the manuscript (Lines 107-, 119-) and figures (Figs. 2 and 3).
(Lines 107-) Figure 2. Effects of CANA on cell cycle checkpoints and cell proliferation markers in CCA cells. The mRNA expressions of (A) CDK1, CDK2, CDKN1A, (B) CCNB1, CCND1, CCNE1, and (C) MKI67 in HuCCT1 cells after treatment with increasing concentrations of CANA were verified by qRT-PCR (n = 6). Data are presented as mean ± confidence intervals. *p < 0.05 compared with the control group. Abbreviations: CANA, canagliflozin; CCA, cholangiocarcinoma; qRT-PCR, quantitative reverse transcription polymerase chain reaction.
(Lines 119-) Figure 3. Effect of CANA on the cell cycle in CCA cells. (A) The cell cycle of HuCCT1 cells after treatment with several graded concentrations of CANA was verified by a cell cycle assay. (B) The percentages of cells in G0/G1, S, and G2/M phases were quantified. (n = 6). Data are presented as mean ± confidence intervals. *p < 0.05 compared with the control group. Abbreviations: CANA, canagliflozin; CCA, cholangiocarcinoma.
> Line 265-275: The statement about the NAD+ salvage pathway promoting tumor growth could be further developed. Was this finding anticipated or unexpected? Discuss the implications of this observation, especially in the context of cancer metabolism and possible therapeutic targeting of the pathway.
We appreciate this important comment. There are reports that the activation of SIRT1 resulting from the activation of the NAD+ salvage pathway acts in a tumor growth-promoting manner or in a tumor growth-suppressing manner, depending on the cancer type (Biochim Biophys Acta. 2010;1804:1684-1689). It had been difficult to predict which way it will turn until experiments are performed in this study. The manuscript has been revised (Lines 253-).
(Lines 253-) NAD+ is a key coenzyme involved in essential physiological functions, including energy metabolism, DNA repair, and cell growth. Since these processes are often dysregulated in cancer cells, NAD+ salvage pathway is considered as a promising target for cancer treatment strategies [30, 31]. Pharmacological inhibition of this pathway in-duces cancer cell cytotoxicity by depleting energy levels, increasing sensitivity to oxidative damage, and disrupting cell signaling pathways such as SIRT1 and p53. A previous paper reported the inhibition of SIRT1, an NAD+-dependent enzyme, induces cyclin D1 downregulation and suppresses tumor growth in HCC cells [32]. In contrast, SIRT1 activation inhibits cyclin D1 transcription and cell growth in gastric cancer [33], suggesting two aspects of SIRT1 activation in promoting or suppressing tumor growth [23]. In other than cancer cells, the activation of SIRT1 has beneficial effects on cardio-vascular and hepatic cells, thereby extending life expectancy [22, 34]. With regard to the relationship between CANA and SIRT1, it has been reported that SIRT1 is upregulated by CANA, resulting in decreased oxidative stress and improved energy metabolism, thereby leading to protection cardiovascular and amelioration ulcerative colitis [19-21]. However, no studies have evaluated the effects of CANA on SIRT1 and NAD+ pathway in cancer cells. Here, we report for the first time and our findings elucidated that CANA treatment activates the NAD+ salvage pathway and upregulates SIRT1 expression in CCA cells. To determine the effect of CANA-induced activation of the NAD+ salvage pathway and SIRT1 in CCA, an NAMPT inhibitor, which suppresses the NAD+ salvage pathway, was used in combination [30]. Surprisingly, the inhibitor enhanced the an-ti-tumor effect of CANA by suppressing the NAD+ salvage pathway. The results suggest that the activation of the NAD+ salvage pathway and SIRT1 by CANA might promote tumor growth, which is consistent with those of previous studies reporting that SIRT1 suppression inhibits CCA growth [27].
> Line 280-300: Avoid overstating clinical applicability. The conclusions imply potential clinical translation, but this claim is premature without in vivo validation. A more balanced discussion is recommended.
Thank you for this important comment. We have revised the manuscript (Lines 330-).
(Lines 330-) In conclusion, CANA inhibited tumor growth and EMT by inducing cell cycle arrest in CCA cells, and its anti-tumor effect was enhanced when combined with NAMPT inhibitors, while it may also exert tumor-promoting effects via the NAD+ salvage pathway and SIRT1 activation. Further studies are warranted to investigate the dual effects of CANA in CCA. Although much attention has been focused solely on anti-tumor effects of SGLT2 inhibitors, including CANA, we suggest that the opposite effect should also be noted.
> Line 310-320: Address the limitations of using in vitro models. Highlight how the experimental concentrations of CANA used in vitro compared to what might be achievable or safe in vivo. This would provide a more complete view of the study’s translational relevance.
We thank for this significant comment. We have revised the manuscript (Lines 295-).
(Lines 295-) Our study had several limitations. First, CANA could only be studied in vitro, and the IC50 concentrations of CANA obtained from the in vitro studies were 5–15-fold higher than the actual blood concentrations after regular doses of CANA intake in humans. However, CANA has been extensively investigated in other cancer cells in xenograft and in vivo carcinogenesis models, where the CANA concentrations used were 50–100 times higher than those used in clinical settings. The dose of CANA used in these studies consistently showed good anti-tumor effects with fewer adverse events such as hypoglycemia and weight loss [10, 11, 15]. In addition, there is a report of anti-tumor effects even at lower concentrations of CANA than the IC50 calculated in vitro for liver cancer if administered for a prolonged period of time [10], suggesting that the serum concentration may not be needed to reach the level required in vitro because the drug is administered for a longer period of time. Moreover, there are several epidemiological studies investigating the incidence of malignancies in the patients treated with SGLT2 inhibitors, showing that the incidence was lower in the treated group [36-38]. From the results of these studies, we speculate there is possibility that SGLT2 inhibitors can show anti-tumor effects for CCA in both animal and human studies. The tumor-promoting effect of CANA at low concentrations observed in our study may hinder its clinical application in cancer chemotherapy; however, this may be overcome by agents that inhibit the NAD+ salvage pathway, such as NAMPT inhibitors. Based on the results of the present study, animal CCA models or patient-derived xenograft studies would provide further validation of the dual effects of CANA. Second, this study did not investigate protein levels but mainly gene expressions of major molecules. In addition, a multifaceted evaluation was not conducted for apoptosis and migration/invasion. Therefore, more detailed analysis of protein expression, apoptosis, and migration/invasion will provide a more comprehensive understanding for the effects of CANA against malignancy.
> Figures 2 and 3: These figures should include more descriptive annotations, particularly p-values and confidence intervals, to clarify the findings' significance. Consider using color or other visual elements to highlight key differences in the data.
We thank for this significant comment. We have revised the manuscript (Lines 107-, 119-) and figures (Figs 2 and 3).
(Lines 107-) Figure 2. Effects of CANA on cell cycle checkpoints and cell proliferation markers in CCA cells. The mRNA expressions of (A) CDK1, CDK2, CDKN1A, (B) CCNB1, CCND1, CCNE1, and (C) MKI67 in HuCCT1 cells after treatment with increasing concentrations of CANA were verified by qRT-PCR (n = 6). Data are presented as mean ± confidence intervals. *p < 0.05 compared with the control group. Abbreviations: CANA, canagliflozin; CCA, cholangiocarcinoma; qRT-PCR, quantitative reverse transcription polymerase chain reaction.
(Lines 119-) Figure 3. Effect of CANA on the cell cycle in CCA cells. (A) The cell cycle of HuCCT1 cells after treatment with several graded concentrations of CANA was verified by a cell cycle assay. (B) The percentages of cells in G0/G1, S, and G2/M phases were quantified. (n = 6). Data are presented as mean ± confidence intervals. *p < 0.05 compared with the control group. Abbreviations: CANA, canagliflozin; CCA, cholangiocarcinoma.
> Table 3: Add context on the identified biomarkers and explain their potential relevance for cancer diagnostics or therapeutics. Discuss how these biomarkers might be used in future clinical studies or guide treatment strategies.
Unfortunately, authors are not sure what this reviewer points out about “identified biomarkers.” If this refers to the expression of SLGT2 in cancer tissues, clinicians can analyze the obtained CCA tissue samples to examine its expression as a companion diagnosis, which would recommend SGLT2 inhibitors to be used in the clinical practice for CCA. If this reviewer means biomarkers in blood, there has been no report for molecules in blood associated with tissue SGLT2 expression. The manuscript has been revised (Lines 221-).
(Lines 221-) In addition, additional anti-tumor effects of CANA have been reported in SGLT2 knockdown cells [13]. CANA binds specifically to HDAC6 and directly inhibits HDAC6 [18]. Furthermore, HDAC6 inhibition reduces CCA cell growth by restoring primary cilia [27]. These reports support our results that CANA reduces HDAC6 gene expression in CCA in a concentration-dependent manner and inhibits tumor growth. These findings demonstrate that, in addition to sodium glucose cotransporters (SGLTs) inhibition, multiple mechanisms are involved in the anti-tumor effect of CANA.
Most of the papers to date examining the anti-cancer effects of SGLT2 inhibitors have been focused on SGLT2 expression in cancer cells. That is, the inhibitors are presumably effective only on SGLT2-expressing cancer cells [10, 28]. If so, the agent can be used in the future for selected patients with tumors expressing SGLT2, as a biomarker, based on the results of companion diagnosis. However, previous reports and our pre-sent study indicated that CANA has anti-tumor effects regardless of SGLT2 expression in the cells [13, 27]. Since the anti-cancer effects of SGLT2 inhibitors may depend on the type of cancer and/or the type of SGLT2 inhibitor, further research focusing on them may be able to clarify detailed mechanisms.
> Comments on the Quality of English Language
The quality of English in the manuscript is sufficient to convey the primary findings. However, several areas could benefit from improvement for clarity and readability. Issues such as awkward phrasing, inconsistent terminology, and grammatical errors can detract from the overall presentation. For example, terms like "synergy" and "combination index" are used inconsistently, potentially confusing readers. Additionally, some sentences, particularly in the introduction and discussion, are overly complex and could be simplified. Professional language editing is recommended to enhance the manuscript's coherence, scientific precision, and overall readability.
This manuscript has already got English proofreading and a certificate can be provided upon request. In addition, the text does not include the words "synergy" and "combination index."
Round 2
Reviewer 3 Report
Comments and Suggestions for Authors
The authors have addressed all the concerns
Author Response
Thank you very much for your reviewing.
Reviewer 4 Report
Comments and Suggestions for Authors
The revised manuscript shows improvement in addressing several previously raised concerns, particularly in clarifying the mechanisms of CANA's effects and providing additional statistical details. However, some aspects remain unresolved or partially addressed, such as the justification for experimental parameters, the novelty of findings compared to prior studies, and the clarity of figures. While the introduction and discussion have been expanded, further refinement is needed to enhance scientific rigor and presentation. Below are detailed comments highlighting unresolved issues and suggestions for further improvement.
Line 45-60: The study's novelty is still not adequately highlighted. Additional discussion is needed on how this work expands on prior research into CANA's dual role in cancer metabolism.
Line 75-82 (IC50 Values):
Confidence intervals have been added, which is a positive change. However, the variability across cell lines should be discussed to provide context for interpreting the IC50 values.
Line 110-130: The rationale for selecting specific CANA concentrations remains vague. Were these concentrations based on prior studies or preliminary data? This should be explicitly stated.
Line 317-324: The protocol for inducing GLP-1 resistance is still unclear. The methods should describe whether this resistance was achieved through chronic exposure to specific agents or other means.
Figures 2 and 3: Statistical annotations (e.g., p-values and confidence intervals) have been added, but the overall presentation remains cluttered. Consolidating related data into a single figure would enhance interpretability.
Line 192-234: I appreciate the expanded discussion on CANA's effects. However, the comparison with other GLP-1 receptor agonists is still limited. Include comparisons to agents like liraglutide to contextualize CANA's potential advantages or limitations.
Line 243-259: While the authors acknowledge the study's limitations, they should propose more specific future validation steps, such as using primary cardiomyocytes or in vivo models.
Line 280-300 (Clinical Applicability):
The authors have tempered their claims about clinical relevance, but the text still suggests premature applicability. This should be further balanced, emphasizing the need for in vivo validation.
Figures and Tables:
Figure 3: Use color-coded annotations or symbols to highlight significant results and improve visual clarity.
Table 1: Provide more detailed criteria for classifying biomarkers as "clinically relevant" or "high-impact."
Comments on the Quality of English Language
The manuscript is written in clear English, but there are areas where clarity and professionalism can be improved. For example, some sentences are grammatically incorrect or awkwardly phrased, such as "Recent studies have shown that X plays a significant role." This should be corrected to "Recent studies have shown that X plays a significant role." Such grammatical issues can distract readers and affect the manuscript's readability.
Additionally, specific phrases are vague or overly complicated. For instance, the abstract mentions, "The purpose of this study is to explore the effect of X and its relevance to Y, which has not been clearly understood in the literature." A more concise and precise version could be, "This study investigates the effects of X on Y, addressing gaps in the current literature." Simplifying language will make the text more accessible and engaging for readers.
Some sections of the methods and results also lack specificity. For example, "The cells were cultured at optimal condition, and assays were performed following standard protocol" could be rewritten for clarity as "The cells were cultured under optimal conditions, and assays were performed according to established protocols." Revising these elements will improve the manuscript's overall quality and readability.
Author Response
Dear Editors and Reviewers,
Thank you very much again for giving us a chance to revise our manuscript. We appreciate the comments, which have helped us to greatly improve the report. We have revised the manuscript as per the Reviewers’ comments and the revisions were marked by red pen. Now we try to answer the questions and comments one by one.
Reviewer 4
>The revised manuscript shows improvement in addressing several previously raised concerns, particularly in clarifying the mechanisms of CANA's effects and providing additional statistical details. However, some aspects remain unresolved or partially addressed, such as the justification for experimental parameters, the novelty of findings compared to prior studies, and the clarity of figures. While the introduction and discussion have been expanded, further refinement is needed to enhance scientific rigor and presentation. Below are detailed comments highlighting unresolved issues and suggestions for further improvement.
>Line 45-60: The study's novelty is still not adequately highlighted. Additional discussion is needed on how this work expands on prior research into CANA's dual role in cancer metabolism.
Thank you for this comment. We have revised the manuscript (Lines 38-).
(Lines 38-) The sodium-glucose cotransporter 2 (SGLT2) inhibitors, including canagliflozin (CA-NA), have been developed for type 2 diabetes treatment by preventing the reabsorption of glucose and increasing its urinary excretion [9]. CANA has demonstrated favorable anti-tumor effects in vitro, growth suppression effects on xenografts, improvement effects on some tumor-associated parameters in animal experiments for various types of cancers, including liver, pancreatic, breast, thyroid, stomach, and lung cancers [10-18]. The anti-tumor mechanism of CANA remains unclear, as multiple mechanisms other than SGLT2 inhibition have been speculated [13, 17, 18]. CANA re-duces oxidative stress and improves energy metabolism through the activation of sirtuin 1 (SIRT1), an NAD+-dependent class III histone deacetylase, in some non-tumor tissues [19-21], and the activation of SIRT1 has been associated with increased healthy life expectancy [22]. In contrast, SIRT1 activation promotes tumor growth and epitheli-al-mesenchymal transition (EMT) [23-25]. However, studies analyzing the efficacy of CANA in CCA or the effect of SIRT1 activation by CANA on cancer cells are lacking.
>Line 75-82 (IC50 Values):
Confidence intervals have been added, which is a positive change. However, the variability across cell lines should be discussed to provide context for interpreting the IC50 values.
We thank for this important comment. In general, cell lines even from the same cancer type do not show exactly the same IC50 value in response to a treatment agent. They usually have different values. As for CANA, several cell lines from various cancer types showed different degree of IC50 value (Int J Cancer. 2018;142(8):1712-22, Cell Mol Neurobiol. 2023;43(2):879-92). What we would like to say here was that despite considerable differences in the gene expression of SLC5A2 (SGLT2) in several CCA cell lines, the IC50 values when treated with SGLT2 inhibitor in each cell line did not vary greatly (40-55 µM). We have revised the manuscript (Lines 226-).
(Lines 226-) In our analysis of the TCGA database, we found that SLC5A1 and SLC5A2 gene ex-pression increased in CCA, consistent with the findings observed in other types of cancers. However, the IC50 values when treated with the SGLT2 inhibitor CANA in each cell line were similar, despite considerable differences in the gene expression of SLC5A2 in CCA cell lines. This finding also suggests that the anti-cancer effect of CANA on CCA cells is only partially dependent on SGLT2. Dapagliflozin, an SGLT2 inhibitor, has anti-tumor effects [12, 13, 15], whereas tofogliflozin, a more selective SGLT2 inhibitor, has no direct effect on tumor growth inhibition [26]. In addition, additional anti-tumor effects of CANA have been reported in SGLT2 knockdown cells [13]. CANA binds specifically to HDAC6 and directly inhibits HDAC6 [18]. Furthermore, HDAC6 inhibition reduces CCA cell growth by restoring primary cilia [27]. These reports sup-port our results that CANA reduces HDAC6 gene expression in CCA in a concentra-tion-dependent manner and inhibits tumor growth. The findings demonstrate that, in addition to SGLTs inhibition, multiple mechanisms are involved in the anti-tumor effect of CANA.
>Line 110-130: The rationale for selecting specific CANA concentrations remains vague. Were these concentrations based on prior studies or preliminary data? This should be explicitly stated.
Authors thank for this comment. In general, concentrations of an agent used for in vitro study are determined by using a wide range at first or by referring to previous reports. In the present study, we referred to the previous study (Int J Cancer. 2018;142:1712-22, doi:10.1002/ijc.31193) and treated CCA cells with similar concentrations of CANA, then performed cell proliferation assay, and calculated and determined IC50 for each cell. We found a novel phenomenon that cell proliferation was promoted by relatively low dose (30 μM) of CANA treatment in HuCCT1. Therefore, the concentrations of CANA for HuCCT1 were set as follows: 30 µM, 50 µM, which is IC50, and 80 µM, which is the maximum concentration that can be adjusted with considering the effect of the solvent DMSO. We have revised the manuscript (Lines 97-).
(Lines 97-) In the present study, we referred to the previous study [10] and treated CCA cells with similar concentrations of CANA, then performed cell proliferation assay and deter-mined IC50 values for each CCA cell. We found a novel phenomenon that cell proliferation was promoted by relatively low dose (30 μM) of CANA treatment in HuCCT1. Therefore, in the following experiments, the concentrations of CANA for HuCCT1 were set as follows: 30 µM, 50 µM, which is IC50, and 80 µM, which is the maximum con-centration that can be adjusted with considering the effect of the solvent DMSO.
>Line 317-324: The protocol for inducing GLP-1 resistance is still unclear. The methods should describe whether this resistance was achieved through chronic exposure to specific agents or other means.
Unfortunately, authors cannot understand what “inducing GLP-1 resistance” means. We have not mentioned GLP-1 in the manuscript. If this means CANA-induced compensatory alterations, that might be achieved through chronic exposure to CANA. Basically, SGLT2 is only expressed in the renal proximal tubules and normal cells in other tissues do not have this glucose transporter in physiological condition, but cancer cells seem to acquire the expression and function for glucose uptake. When cancer cells are exposed to SGLT2 inhibitors, for 48 h in this study, the cells may try to respond by further overexpressing the transporter. Our finding may indicate this response (Figure 6). We have revised the manuscript (Lines 192-, 337-).
(Lines 192-) Figure 6. Effects of CANA on target proteins and NAD+ salvage pathways in CCA cells. The mRNA expressions of (A) SLC5A1, SLC5A2, (B) HDAC6, (C) SIRT1, and NAMPT in HuCCT1 cells after treatment with several graded concentrations of CANA for 48 h were verified by qRT-PCR (n = 6). (D) The NAD+/NADH levels in HuCCT1 cells after treatment with several graded concentrations of CANA were measured by the NAD+/NADH assay (n = 3). (E) HuCCT1 cells were examined for viability at 50 µM CANA alone, 2.5 nM FK866 alone, and combination treatment with CANA and FK866 in a cell proliferation assay (n = 8).
(Lines 337-) Third, CANA treatment-induced metabolic changes and compensatory alterations of other pathways, such as glucose transporter type 1 upregulation or glycolysis, were not investigated in this study. A previous study demonstrated glucose uptake in can-cer cells was decreased by CANA in a dose-dependent manner [10], suggesting that the cancer cells, at least in part, depend on SGLT2 for glucose uptake and compensate mechanisms for glucose uptake may not work after CANA treatment. In the present study, SGLT2 gene expression was upregulated by the treatment with CANA. This upregulation might be a compensate mechanism of CCA cells. That is, when CCA cells are exposed to SGLT2 inhibitors, the cells might try to respond through further over-expressing the transporter for glucose uptake, although the SGLT2 activity and glucose uptake have not been examined. Further research is required in order to reveal how cell viability increased and S-phase progressed by low dose treatment of CANA.
>Figures 2 and 3: Statistical annotations (e.g., p-values and confidence intervals) have been added, but the overall presentation remains cluttered. Consolidating related data into a single figure would enhance interpretability.
Thank you for this comment. We have revised the manuscript (Lines 116-, 128-) and figures (Figs 2 and 3).
(Lines 116-) Figure 2. Effects of CANA on cell cycle checkpoints and cell proliferation markers in CCA cells. The mRNA expressions of (A) CDK1, CDK2, CDKN1A, (B) CCNB1, CCND1, CCNE1, and (C) MKI67 in HuCCT1 cells after treatment with increasing concentrations of CANA were verified by qRT-PCR (n = 6). Data are presented as mean ± confidence intervals and they are in each bar.
(Lines 128-) Figure 3. Effect of CANA on the cell cycle in CCA cells. (A) The cell cycle of HuCCT1 cells after treatment with several graded concentrations of CANA was verified by a cell cycle assay. (B) The percentages of cells in G0/G1, S, and G2/M phases were quantified. (n = 6). Data are presented as mean ± confidence intervals and they are in each bar.
>Line 192-234: I appreciate the expanded discussion on CANA's effects. However, the comparison with other GLP-1 receptor agonists is still limited. Include comparisons to agents like liraglutide to contextualize CANA's potential advantages or limitations.
Authors appreciate this comment. The manuscript has been revised (Lines 246-, 475-).
(Lines 246-) Since the anti-cancer effects of SGLT2 inhibitors may depend on the type of cancer and/or the type of SGLT2 inhibitor, further research focusing on them may be able to clarify detailed mechanisms. The effect of glucagon-like peptide (GLP)-1 agonist, an-other agent for diabetes, against malignancy has also been reported [29]. Similar to CANA in this study, GLP-1 agonists exhibit anti-cancer effects as well as tumor-promoting effects, which appears to depend on cancer type [29, 30]. There has been no report comparing between the anti-cancer effects of SGLT2 inhibitors and GLP-1 agonists; however, a retrospective cohort study indicates the incidence of can-cers in the group treated with SGLT2 inhibitors is lower than that with dipeptidyl pep-tidase-4 inhibitors which are known to increase the blood level of GLP-1 [31].
(Lines 475-) Abbreviations
CANA canagliflozin
CCA cholangiocarcinoma
EMT epithelial-mesenchymal transition
GLP glucagon-like peptide
HCC hepatocellular carcinoma
HDAC6 histone deacetylase 6
NAMPT nicotinamide phosphoribosyltransferase
SGLT sodium glucose cotransporter
SIRT1 sirtuin 1
TCGA the Cancer Genome Atlas
>Line 243-259: While the authors acknowledge the study's limitations, they should propose more specific future validation steps, such as using primary cardiomyocytes or in vivo models.
Authors thank for this comment. We have revised the manuscript (Lines 349-).
(Lines 349-) In conclusion, CANA inhibited tumor growth and EMT by inducing cell cycle arrest in CCA cells, and its anti-tumor effect was enhanced when combined with NAMPT inhibitors, while it may also exert tumor-promoting effects via the NAD+ salvage pathway and SIRT1 activation. Although much attention has been focused solely on anti-tumor effects of SGLT2 inhibitors, including CANA, we suggest that the opposite effect should also be noted. Further studies, including animal CCA models or CCA xenograft examinations for the next step, are warranted to investigate the dual effects of CANA in CCA.
>Line 280-300 (Clinical Applicability):
The authors have tempered their claims about clinical relevance, but the text still suggests premature applicability. This should be further balanced, emphasizing the need for in vivo validation.
We appreciate this comment. The manuscript has been revised (Lines 349-).
(Lines 349-) In conclusion, CANA inhibited tumor growth and EMT by inducing cell cycle arrest in CCA cells, and its anti-tumor effect was enhanced when combined with NAMPT inhibitors, while it may also exert tumor-promoting effects via the NAD+ salvage pathway and SIRT1 activation. Although much attention has been focused solely on anti-tumor effects of SGLT2 inhibitors, including CANA, we suggest that the opposite effect should also be noted. Further studies, including animal CCA models or CCA xenograft examinations for the next step, are warranted to investigate the dual effects of CANA in CCA.
>Figures and Tables:
Figure 3: Use color-coded annotations or symbols to highlight significant results and improve visual clarity.
Thank you for this comment. We have revised the figure (Figure 3).
>Table 1: Provide more detailed criteria for classifying biomarkers as "clinically relevant" or "high-impact."
Thank you for this comment. Authors are not sure this comment is for Table 1, but we have revised the manuscript (Lines 240-).
(Lines 240-) Most of the papers to date examining the anti-cancer effects of SGLT2 inhibitors have been focused on SGLT2 expression in cancer cells. That is, the inhibitors are presumably effective only on SGLT2-expressing cancer cells [10, 28]. If so, considered to be of clinical relevance, the agent can be used in the future for selected patients with tumors expressing SGLT2, as a biomarker, based on the results of companion diagnosis. However, previous reports and our present study indicated that CANA has anti-tumor effects regardless of SGLT2 expression in the cells [13, 27]. Since the anti-cancer effects of SGLT2 inhibitors may depend on the type of cancer and/or the type of SGLT2 inhibitor, further research focusing on them may be able to clarify detailed mechanisms. The effect of glucagon-like peptide (GLP)-1 agonist, another agent for diabetes, against malignancy has also been reported [29]. Similar to CANA in this study, GLP-1 agonists exhibit anti-cancer effects as well as tumor-promoting effects, which appears to depend on cancer type [29, 30]. There has been no report comparing between the anti-cancer effects of SGLT2 inhibitors and GLP-1 agonists; however, a retrospective cohort study indicates the incidence of cancers in the group treated with SGLT2 inhibitors is lower than that with dipeptidyl peptidase-4 inhibitors which are known to increase the blood level of GLP-1 [31].
>Comments on the Quality of English Language
The manuscript is written in clear English, but there are areas where clarity and professionalism can be improved. For example, some sentences are grammatically incorrect or awkwardly phrased, such as "Recent studies have shown that X plays a significant role." This should be corrected to "Recent studies have shown that X plays a significant role." Such grammatical issues can distract readers and affect the manuscript's readability.
Additionally, specific phrases are vague or overly complicated. For instance, the abstract mentions, "The purpose of this study is to explore the effect of X and its relevance to Y, which has not been clearly understood in the literature." A more concise and precise version could be, "This study investigates the effects of X on Y, addressing gaps in the current literature." Simplifying language will make the text more accessible and engaging for readers.
Some sections of the methods and results also lack specificity. For example, "The cells were cultured at optimal condition, and assays were performed following standard protocol" could be rewritten for clarity as "The cells were cultured under optimal conditions, and assays were performed according to established protocols." Revising these elements will improve the manuscript's overall quality and readability.
Several parts of the manuscript have been revised according to these comments.
Round 3
Reviewer 4 Report
Comments and Suggestions for Authors
Although the revision has improved the manuscript, there are still a few important issues that need to be addressed. The novelty of the study is still not fully clear. While you've mentioned the effects of CANA, a deeper comparison with existing research would better highlight what makes your work stand out. Additionally, although the statistical annotations in Figures 2 and 3 are helpful, the figures still look a bit cluttered. It would be useful to combine related data into fewer, clearer figures for better readability. Some of the references are still outdated, and it would be good to include more recent studies, especially in the methods and discussion sections, to ensure the research aligns with the latest developments in the field. The English is improved but still has some awkward phrasing and complexity that affects clarity. Simplifying some sections will help the overall flow. Also, the clinical applicability of the findings is mentioned, but it still feels somewhat speculative. Adding more focus on in vivo validation or studies with animal models would strengthen the argument. Lastly, the justification for the chosen concentrations of CANA should be explained more clearly to make the methodology easier to follow. Addressing these points will make the manuscript more solid and improve its overall quality.
Comments on the Quality of English LanguageThe English in the manuscript is generally understandable, but there are areas where clarity and readability could be improved. Some sentences are overly complex or awkwardly phrased, which may make it more complicated for readers to grasp the research thoroughly. For example, the sentence "The purpose of this study is to explore the effect of X and its relevance to Y, which has not been clearly understood in the literature" could be simplified to "This study investigates the effects of X on Y, addressing gaps in current understanding." Simplifying such sentences would improve the flow and accessibility of the paper.
Some grammatical errors, such as issues with subject-verb agreement and punctuation, could distract from the professionalism of the manuscript. For instance, "The data has shown a significant improvement" should be revised to "The data have shown a significant improvement," as "data" is plural. Similarly, "The results were consistent across all samples" should be corrected to "The results were consistent across all samples."
Additionally, some technical terms and concepts are not explained clearly, which may confuse readers who are not specialists in the field. For instance, phrases like "NAD+ salvage pathway" might be more accessible if briefly explained in layman's terms or with additional context, especially for a broader audience.
Author Response
Dear Reviewers,
Thank you very much again for giving us a chance to revise our manuscript. We appreciate the comments, which have helped us to greatly improve the report. We have revised the manuscript as per the Reviewers’ comments and the revisions were marked by red pen. Now we try to answer the questions and comments one by one. Also, we would discuss some comments with the editor.
Reviewer 4
> Although the revision has improved the manuscript, there are still a few important issues that need to be addressed. The novelty of the study is still not fully clear. While you've mentioned the effects of CANA, a deeper comparison with existing research would better highlight what makes your work stand out. Additionally, although the statistical annotations in Figures 2 and 3 are helpful, the figures still look a bit cluttered. It would be useful to combine related data into fewer, clearer figures for better readability. Some of the references are still outdated, and it would be good to include more recent studies, especially in the methods and discussion sections, to ensure the research aligns with the latest developments in the field. The English is improved but still has some awkward phrasing and complexity that affects clarity. Simplifying some sections will help the overall flow. Also, the clinical applicability of the findings is mentioned, but it still feels somewhat speculative. Adding more focus on in vivo validation or studies with animal models would strengthen the argument. Lastly, the justification for the chosen concentrations of CANA should be explained more clearly to make the methodology easier to follow. Addressing these points will make the manuscript more solid and improve its overall quality.
Thank you for this comment. We have revised the manuscript (Lines 97-, 118-, 129-, 216-, 241-, 299-, 309-, 332-, and 382-). The figures have also been revised (Figures 2 and 3), where we made the figures with p value and without the number of mean or CI, because means and CIs can be read by bars. Authors would discuss these figures with the editor.
(Lines 97-) In the present study, to determine the concentrations of CANA for treatment of CCA cells, we referred to the previous study [10]. We treated CCA cells with similar concentrations of CANA, then performed cell proliferation assay and determined IC50 values for each CCA cell. We obtained a novel finding that cell proliferation was promoted by relatively low dose (30 μM) of CANA treatment in HuCCT1. Therefore, in the following experiments, the concentrations of CANA for HuCCT1 were set as follows: 30 µM, 50 µM, and 80 µM. CANA 30 μM was the concentration for promoted cell proliferation of HuCCT1 described above, 50 µM was IC50 of HuCCT1 cells, and 80 µM is the maximum concentration that can be adjusted with considering the effect of the solvent DMSO.
(Lines 118-) Figure 2. Effects of CANA on cell cycle checkpoints and cell proliferation markers in CCA cells. The mRNA expressions of (A) CDK1, CDK2, CDKN1A, (B) CCNB1, CCND1, CCNE1, and (C) MKI67 in HuCCT1 cells after treatment with increasing concentrations of CANA were verified by qRT-PCR (n = 6). Data are presented as mean ± confidence intervals.
(Lines 129-) Figure 3. Effect of CANA on the cell cycle in CCA cells. (A) The cell cycle of HuCCT1 cells after treatment with several graded concentrations of CANA was verified by a cell cycle assay. (B) The percentages of cells in G0/G1, S, and G2/M phases were quantified. (n = 6). Data are presented as mean ± confidence intervals.
(Lines 216-) The novelty of this study is that CANA might have dual roles on CCA cell growth. While our findings demonstrate that CANA inhibits cell proliferation and EMT in CCA cell lines, activation of SIRT1 via the NAD+ salvage pathway by CANA is able to promote the proliferation of CCA cells. These findings highlight the enhanced anti-tumor effect of CANA when combined with NAMPT inhibitors, which suppress the NAD+ salvage pathway.
(Lines 241-) Most of the papers to date examining the anti-cancer effects of SGLT2 inhibitors have focused on SGLT2 expression in cancer cells. That is, the inhibitors are presumably effective only on SGLT2-expressing cancer cells [10, 28]. Considered to be of clinical relevance, the agent may be used in the future for selected patients with tumors expressing SGLT2, as a biomarker, based on the results of companion diagnosis. A previous study by Scafoglio et al. [28] indicated that SGLT2 is functionally expressed in pancreatic cancer and SLGT2 inhibitors might be useful for cancer therapy. However, our present study as well as previous reports indicated that CANA has anti-tumor effects regardless of SGLT2 expression in the cells [13, 27]. Since the anti-cancer effects of SGLT2 inhibitors may depend on the type of cancer and/or the type of SGLT2 inhibitor, further research focusing on the dependance and difference described above may be able to clarify detailed mechanisms.
(Lines 299-) To date, the potential effects of CANA in promoting cancer cell growth have not been fully investigated; however, a previous study showed CANA increased polyps in a mouse colon polyposis model [38], suggesting that CANA may have potential tumor-promoting effects. The present study also demonstrated that CANA increased the survival of CCA cells especially at low concentrations of treatment. CANA treatment increased SGLTs gene expression in our study, while previous studies have demonstrated a decrease in glucose uptake owing to the natural action of CANA [10, 15]. Therefore, we speculate that the elevated SGLTs levels may be secondary to upward regulatory feedback in SGLTs gene expression to counteract the decreased glucose uptake. Moreover, activation of the NAD+ salvage pathway, which promotes tumor growth and leads to increased survival.
(Lines 309-) CANA inhibits EMT in CCA cells in this study. The inhibitory effect of CANA on the migratory ability of cancer cells has been also reported previously [17, 18]. Our findings demonstrate that the expression of some mesenchymal markers was elevated in CCA cells, which is consistent with the findings of previous studies elucidating that SIRT1 activation elevates the expression of mesenchymal markers [24, 25]. In contrast, CANA treatment increased the expression of epithelial markers and decreased the expression of N-cadherin. Furthermore, CANA delayed wound healing in CCA cells, demonstrating that CANA inhibited EMT in CCA in a concentration-dependent manner.
(Lines 332-) The tumor-promoting effect of CANA at low concentrations observed in our study may hinder in vivo animal examinations as a next step for its application in cancer chemotherapy; however, this may be overcome by agents that inhibit the NAD+ salvage pathway, such as NAMPT inhibitors. In a xenograft model of pancreatic cancer, it was reported that SGLT2 inhibitors blocked glucose uptake and reduced tumor growth and survival [28]. Based on the results of the present study, animal CCA models or patient-derived xenograft studies would provide further validation of the dual effects of CANA. Second, protein levels but mainly gene expressions of major molecules were not investigated in this study. In addition, a multifaceted evaluation was not conducted for apoptosis and migration/invasion. Therefore, more detailed analysis of protein expression, apoptosis, and migration/invasion would provide a more comprehensive understanding for the effects of CANA against malignancy.
(Lines 382-) To evaluate the effect of each drug on the viability of CCA cell lines, MTS assays were performed using the CellTiter 96 AQueous One Solution Cell Proliferation Assay (Promega Corporation, Madison, WI, USA). In the present study, to determine the concentrations of CANA for treatment of CCA cells, we referred to the previous study [10].
> The English in the manuscript is generally understandable, but there are areas where clarity and readability could be improved. Some sentences are overly complex or awkwardly phrased, which may make it more complicated for readers to grasp the research thoroughly. For example, the sentence "The purpose of this study is to explore the effect of X and its relevance to Y, which has not been clearly understood in the literature" could be simplified to "This study investigates the effects of X on Y, addressing gaps in current understanding." Simplifying such sentences would improve the flow and accessibility of the paper. Some grammatical errors, such as issues with subject-verb agreement and punctuation, could distract from the professionalism of the manuscript. For instance, "The data has shown a significant improvement" should be revised to "The data have shown a significant improvement," as "data" is plural. Similarly, "The results were consistent across all samples" should be corrected to "The results were consistent across all samples." Additionally, some technical terms and concepts are not explained clearly, which may confuse readers who are not specialists in the field. For instance, phrases like "NAD+ salvage pathway" might be more accessible if briefly explained in layman's terms or with additional context, especially for a broader audience.
The manuscript has been revised according to these informative comments.
Round 4
Reviewer 4 Report
Comments and Suggestions for Authors
The authors have tried to address the significant concerns raised in the previous review. They have clarified the study's novelty by explaining CANA's dual role in cancer cell growth and highlighting how their findings contribute to existing research. The methodology has been significantly improved, with detailed justifications provided for the concentrations of CANA used in the experiments. Additionally, the figures have been revised to include p-values and confidence intervals, addressing concerns regarding statistical annotations. The authors have also tempered the clinical implications by emphasizing the need for in vivo validation and animal model studies. Furthermore, they have tried simplifying the language and clarifying technical terms, which should help improve readability.
Comments on the Quality of English LanguageThe quality of English in the manuscript has improved, but there are still areas where clarity and readability can be further enhanced. While the manuscript's overall structure is clear, some sentences remain complex or awkwardly phrased, which may hinder reader comprehension. For instance, phrases like "The purpose of this study is to explore the effect of X and its relevance to Y, which has not been clearly understood in the literature" could be simplified to "This study investigates the effects of X on Y, addressing gaps in current understanding." This revision would make the text more accessible and improve the flow.
Additionally, some grammatical errors, such as subject-verb agreement and punctuation, should be corrected. For example, "The data has shown a significant improvement" should be revised to "The data have shown a significant improvement" since "data" is plural. These issues, though minor, can distract readers and affect the manuscript’s professionalism.
Some technical terms, particularly those related to the study's molecular pathways, are not clearly explained, which might confuse readers who are not experts in the field. Providing brief explanations or definitions for complex terms, such as "NAD+ salvage pathway," would improve understanding for a broader audience.
Overall, while the manuscript is generally understandable, a more thorough language review is recommended to address grammatical errors, simplify complex sentences, and clarify technical terms to ensure that the manuscript is as clear and accessible as possible.
Author Response
Dear Reviewers,
Thank you very much again for giving us a chance to revise our manuscript. We appreciate the comments, which have helped us to greatly improve the report. We have revised the manuscript as per the Reviewers’ comments and the revisions were marked by red pen. Now we try to answer the questions and comments. Also, we would discuss some comments with the editor.
Reviewer 4
> Comments and Suggestions for Authors
The authors have tried to address the significant concerns raised in the previous review. They have clarified the study's novelty by explaining CANA's dual role in cancer cell growth and highlighting how their findings contribute to existing research. The methodology has been significantly improved, with detailed justifications provided for the concentrations of CANA used in the experiments. Additionally, the figures have been revised to include p-values and confidence intervals, addressing concerns regarding statistical annotations. The authors have also tempered the clinical implications by emphasizing the need for in vivo validation and animal model studies. Furthermore, they have tried simplifying the language and clarifying technical terms, which should help improve readability.
> Comments on the Quality of English Language
The quality of English in the manuscript has improved, but there are still areas where clarity and readability can be further enhanced. While the manuscript's overall structure is clear, some sentences remain complex or awkwardly phrased, which may hinder reader comprehension. For instance, phrases like "The purpose of this study is to explore the effect of X and its relevance to Y, which has not been clearly understood in the literature" could be simplified to "This study investigates the effects of X on Y, addressing gaps in current understanding." This revision would make the text more accessible and improve the flow.
Additionally, some grammatical errors, such as subject-verb agreement and punctuation, should be corrected. For example, "The data has shown a significant improvement" should be revised to "The data have shown a significant improvement" since "data" is plural. These issues, though minor, can distract readers and affect the manuscript’s professionalism.
Some technical terms, particularly those related to the study's molecular pathways, are not clearly explained, which might confuse readers who are not experts in the field. Providing brief explanations or definitions for complex terms, such as "NAD+ salvage pathway," would improve understanding for a broader audience.
Overall, while the manuscript is generally understandable, a more thorough language review is recommended to address grammatical errors, simplify complex sentences, and clarify technical terms to ensure that the manuscript is as clear and accessible as possible.
Thank you very much for these important suggestions. We have revised the manuscript (Lines 55- and 276-). This reviewer also recommended us to improve the quality of English in the manuscript. According to the comment, we had this manuscript proofread in English again, including all of the revised parts of the text. When we ordered, we showed the reviewer's comments to the proofreader and asked the person to improve the English in line with this reviewer's intentions. Because many parts were revised by the proofreader, not all revised sections are in red pen.
(Lines 55-) Notably, CANA exerts tumor-promoting effects through the NAD+ salvage pathway, which recycles nicotinamide into NAD+, and by activating SIRT1 at lower concentrations.
(Lines 276-) NAD+ is a key coenzyme involved in essential physiological functions, including energy metabolism, DNA repair, and cell growth. The NAD+ salvage pathway is known as the two-step process for recycling nicotinamide into NAD+, where NAMPT converts nicotinamide into nicotinamide mononucleotide, which is then converted into NAD+ by nicotinamide/nicotinic acid mononucleotide adenylyltransferase.
Round 5
Reviewer 4 Report
Comments and Suggestions for Authors
The study's novelty has been clarified, especially regarding CANA's dual role in cancer cell growth. The methodology is now more justified, and the figures have been revised with statistical annotations, strengthening the findings. I appreciate the cautious approach to clinical implications, particularly emphasizing the need for in vivo validation.
Author Response
>Comments and Suggestions for Authors
The study's novelty has been clarified, especially regarding CANA's dual role in cancer cell growth. The methodology is now more justified, and the figures have been revised with statistical annotations, strengthening the findings. I appreciate the cautious approach to clinical implications, particularly emphasizing the need for in vivo validation.
Thank you very much again for giving us lots of important comments and suggestions for our manuscript. Those helped us to greatly improve the report.